# DNA methyltransferase DNMT3a contributes to neuropathic pain by repressing *Kcna2* in primary afferent neurons

Jian-Yuan Zhao[1,2,*], Lingli Liang[1,*], Xiyao Gu[1,*], Zhisong Li[1,3,*], Shaogen Wu[1], Linlin Sun[1], Fidelis E. Atianjoh[1,4], Jian Feng[5], Kai Mo[1], Shushan Jia[1], Brianna Marie Lutz[1], Alex Bekker[1], Eric J. Nestler[5] & Yuan-Xiang Tao[1,3,6]

Nerve injury induces changes in gene transcription in dorsal root ganglion (DRG) neurons, which may contribute to nerve injury-induced neuropathic pain. DNA methylation represses gene expression. Here, we report that peripheral nerve injury increases expression of the DNA methyltransferase DNMT3a in the injured DRG neurons via the activation of the transcription factor octamer transcription factor 1. Blocking this increase prevents nerve injury-induced methylation of the voltage-dependent potassium (Kv) channel subunit *Kcna2* promoter region and rescues *Kcna2* expression in the injured DRG and attenuates neuropathic pain. Conversely, in the absence of nerve injury, mimicking this increase reduces the *Kcna2* promoter activity, diminishes *Kcna2* expression, decreases Kv current, increases excitability in DRG neurons and leads to spinal cord central sensitization and neuropathic pain symptoms. These findings suggest that DNMT3a may contribute to neuropathic pain by repressing *Kcna2* expression in the DRG.

[1] Department of Anesthesiology, New Jersey Medical School, Rutgers, The State University of New Jersey, Newark, New Jersey 07103, USA. [2] State Key Laboratory of Genetic Engineering, School of Life Sciences, Fudan University, Shanghai 200438, China. [3] Department of Anesthesiology, The First Affiliated Hospital of Zhengzhou University, Zhengzhou 450052, Henan, China. [4] Department of Anatomy, College of Medicine, Howard University, Washington DC 20059, USA. [5] Fishberg Department of Neuroscience and Friedman Brain Institute, Icahn School of Medicine at Mount Sinai, New York, New York 10029, USA. [6] Departments of Cell Biology & Molecular Medicine and Physiology, Pharmacology & Neuroscience, New Jersey Medical School, Rutgers, The State University of New Jersey, Newark, New Jersey 07103, USA. * These authors contributed equally to this work. Correspondence and requests for materials should be addressed to Y.-X.T. (email: yuanxiang.tao@njms.rutgers.edu).

Neuropathic pain resulting from peripheral nerve injury is a long-term, debilitating condition affecting the quality of life of over 50 million people in the United States alone[1]. It is characterized by spontaneous ongoing pain or intermittent burning pain, an exaggerated response to painful stimuli (hyperalgesia), and pain in response to normally innocuous stimuli (allodynia). One of the primary causes of these pain hypersensitivities is hyperexcitability and abnormal ectopic firing that arises in neuromas at the sites of peripheral nerve injury and the primary sensory neurons of dorsal root ganglia (DRG)[2–5]. Voltage-gated potassium (Kv) channels that control resting membrane potentials and the excitability of DRG neurons are considered to be key players in neuropathic pain genesis[6–12]. Peripheral nerve injury leads to a significant decrease expression of voltage-gated potassium (Kv) channels, such as Kv1.2, encoded by Kcna2, at both transcriptional and translational levels in the injured DRG, which participate in neuropathic pain genesis[6–9,13–18].

DNA methylation, one type of epigenetic modification, represses gene expression[19–21]. DNA methylation is caused primarily by a family of DNA methyltransferases (DNMTs) including DNMT1, DNMT3a and DNMT3b (refs 19,22,23). Conventionally, DNMT1 acts as the primary maintenance methyltransferase to keep the methylation of DNA that is already established at the genome, whereas DNMT3a and DNMT3b are classified as *de novo* methyltransferases to reversibly methylate unmethylated DNA[22,23]. DNA methylation represses gene transcription through several mechanisms including physically blocking the binding of transcription factors and/or functioning as docking sites for transcriptional repressors/corepressors[19–24]. How DNMTs contribute to neuropathic pain genesis is still elusive, although other epigenetic mechanisms (for example, histone modifications and non-coding RNAs) have recently been implicated in rodent models of neuropathic pain[19,25–33].

We report here that the *de novo* methyltransferase DNMT3a, but not DNMT3b, is significantly increased in the injured DRG neurons after peripheral nerve injury. This increase contributes to injury-induced epigenetic silencing of the *Kcna2* gene in the DRG. Given that *Kcna2* is a key player in neuropathic pain genesis[7–9], DNMT3a likely acts as an endogenous contributor to neuropathic pain development in the injured DRG.

## Results

### DNMT3a is increased in DRG after peripheral nerve injury.
To explore the potential role of DRG DNMT3a in neuropathic pain, we first analysed its distribution pattern in the rat DRG. Using double labelling for DNMT3a and NeuN (a specific neuronal marker) or triple labelling for DNMT3a, glutamine synthetase (GS, a marker for satellite glial cells) and 4′, 6-diamidino-2-phenylindole (DAPI, a marker for cellular nuclei), we found that DNMT3a co-expressed with NeuN in cellular nuclei (Fig. 1a) and was not detected in the cellular nuclei (labelled by DAPI) of GS-labelled cells (Fig. 1b), indicating that DNMT3a is expressed exclusively in the neurons of DRG. Approximately 50.8% of NeuN-labelled neurons (488 of 963) were positive for DNMT3a. A cross-sectional area analysis of neuronal somata found that ~21.6% of DNMT3a-positive neurons are small ($<600\,\mu m^2$ in area), 48.2% are medium ($600–1,200\,\mu m^2$ in area) and 30.2% are large ($>1,200\,\mu m^2$ in area) (Fig. 1c). Consistently, about 34.6% of DNMT3a-positive neurons were labelled by calcitonin gene-related peptide (CGRP, a marker for small DRG peptidergic neurons, Fig. 1d), 23.6% by isolectin B4 (IB4, a marker for small non-peptidergic neurons, Fig. 1e) and 48.2% by neurofilament-200

(NF200, a marker for medium/large cells and myelinated Aβ-fibres, Fig. 1f).

We further examined whether peripheral nerve injury altered DNMT3a expression in two pain-related regions, DRG and spinal cord. The level of DNMT3a protein significantly increased in the ipsilateral (injured) L5 DRG on days 3 and 7, but not on day 14, after unilateral fifth lumbar (L5) spinal nerve ligation (SNL) (Fig. 2a). The number of DNMT3a-positive neurons in the ipsilateral L5 DRG on days 3 and 7 post-SNL also increased by 1.23-fold and 1.24-fold, respectively, compared with the corresponding sham group (Fig. 2b,c). Furthermore, *Dnmt3a* mRNA levels on days 3 and 7 post-SNL were greater than those in the corresponding sham group (Fig. 2d). Interestingly, the expression of another *de novo* methyltransferase DNMT3b protein did not change in the injured DRG during the observation period (Fig. 2a). As expected, sham surgery did not alter the basal expression of both DNMT3a and DNMT3b proteins in the ipsilateral L5 DRG following surgery (Fig. 2a). Neither SNL nor sham surgery changed the basal expression of DNMT3a and DNMT3b proteins in the contralateral L5 DRG, the ipsilateral L4 (intact) DRG or ipsilateral L5 spinal cord dorsal horn (Supplementary Fig. 1a–c). Similar results were found after chronic constriction injury (CCI) of unilateral sciatic nerve, another preclinical neuropathic pain model[34]. The levels of *Dnmt3a* (but not *Dnmt3b*) mRNA and protein on day 7 post-CCI were higher than those on day 7 post-sham surgery (Fig. 2e,f). To check whether peripheral chronic inflammation also changed DNMT3a and DNMT3b expression in the DRG, we injected complete Freund's adjuvant (CFA) unilaterally into the plantar side of the hindpaw[35,36]. Unexpectedly, the amounts of DNMT3a and DNMT3b proteins were not significantly altered in the ipsilateral L4 and L5 DRG from 2 h to 7 days post-CFA injection (Supplementary Fig. 1d). The level of DNA methylation is also controlled by ten-eleven translocation methylcytosine dioxygenases (TETs), which cause oxidation of methylated DNA and result in demethylation[37]. We found that the amounts of TET1, TET2 and TET3 did not change in the ipsilateral L5 DRG on days 3 and 7 post-SNL compared to naive animals (Supplementary Fig. 1e). Our findings indicate that transactivation of *Dnmt3a* mRNA and subsequent production of DNMT3a protein occur exclusively in the injured DRG neurons, specifically in the early period post injury.

### OCT1 promotes DRG *Dnmt3a* gene activity after SNL.
Next, we examined how DRG *Dnmt3a* transcription is upregulated after peripheral nerve injury. Using the online software TFSEARCH, we identified a consensus binding motif ($_{-499}$AATGACAT$_{-442}$) for octamer transcription factor 1 (OCT1) in the promoter region of the *Dnmt3a* gene. A chromatin immunoprecipitation (ChIP) assay revealed that a fragment within the *Dnmt3a* gene promoter that included the above binding motif could be amplified from the complex immunoprecipitated with OCT1 antibody (but not normal serum) in nuclear fractions from the DRG of sham animals (Fig. 3a), indicating the existence of specific binding of OCT1 to the *Dnmt3a* gene in DRG. SNL significantly increased the binding activity, as shown by a 2.75-fold increase in band density in the ipsilateral L5 DRG on day 7 post-SNL compared to that on day 7 post-sham surgery (Fig. 3a). The increased binding activity may be due to time-dependent upregulation of OCT1 protein in the ipsilateral L5 DRG after SNL (Fig. 3b). As expected, sham surgery did not alter the basal expression of OCT1 in the ipsilateral L5 DRG (Fig. 3b). Neither SNL nor sham surgery altered the basal expression of OCT1 in the contralateral L5 DRG and ipsilateral L4 DRG (Supplementary Fig. 1f).

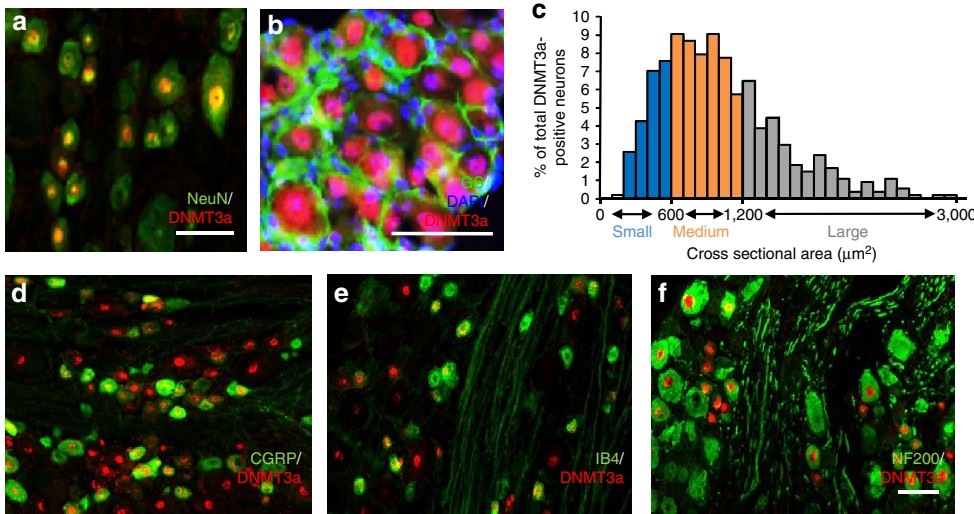

**Figure 1 | Distribution of DNMT3a protein in lumbar dorsal root ganglia of naive rats.** (**a,b**) DNMT3a is co-expressed exclusively with NeuN in cellular nuclei (**a**) and undetected in cellular nuclei (labelled by 4′, 6-diamidino-2-phenylindole (DAPI)) of glutamine synthetase (GS)-labelled cells (**b**). (**c**) Distribution of DNMT3a-positive somata: large, 30.2%; medium, 48.2%; small, 21.6%. (**d-f**) DNMT3a-positive neurons were labelled by calcitonin gene-related peptide (CGRP; **d**), isolectin B4 (IB4; **e**) or neurofilament-200 (NF200; **f**). $n = 5$ rats. Scale bars, 50 μm.

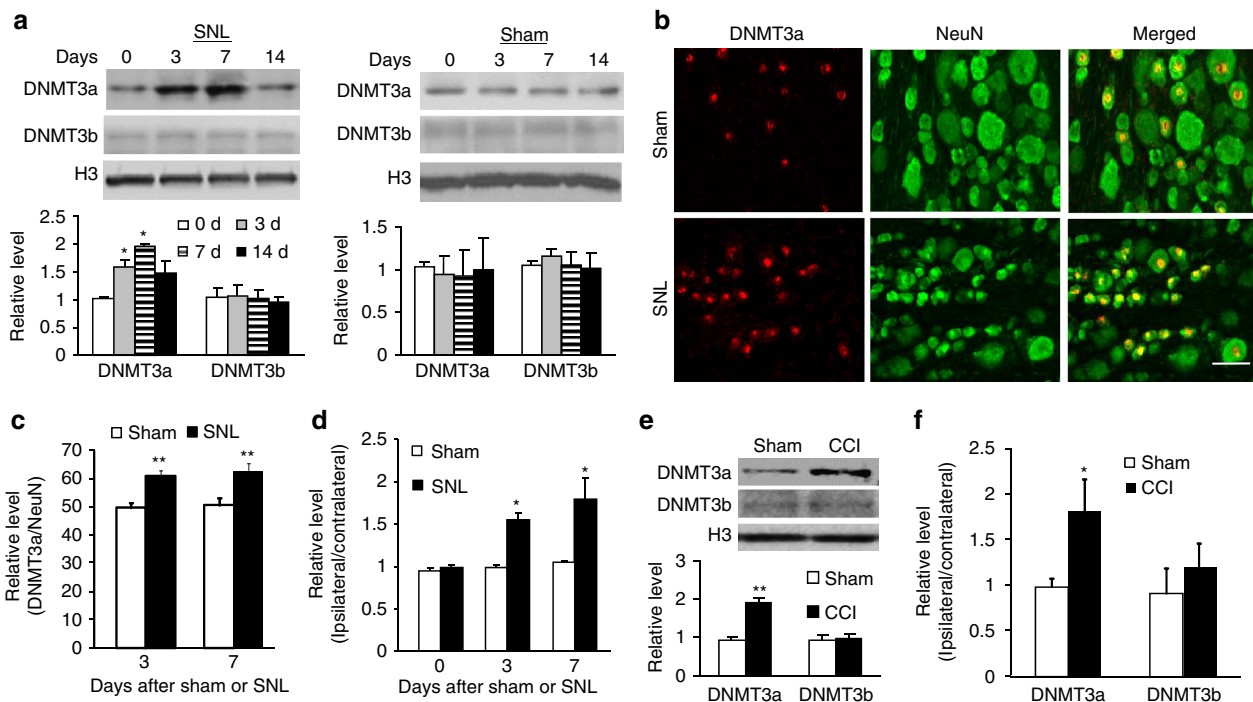

**Figure 2 | Nerve injury-induced increases in *Dnmt3a* mRNA and DNMT3a protein levels in the injured DRG.** (**a**) DNMT3a and DNMT3b protein expression in the ipsilateral L5 DRG after SNL or sham surgery. $n = 6$ rats/time point. One-way ANOVA followed by *post hoc* Tukey test, $F_{time}$ (3, 15) = 8.09 for DNMT3a and $F_{time}$ (3, 15) = 0.07 for DNMT3b. *$P < 0.05$ versus the corresponding control group (0 day). Full-length blots are presented in Supplementary Fig. 6. (**b,c**) Neurons labelled by DNMT3a and NeuN in the ipsilateral L5 DRG on days 3 (**c**) and 7 (**b,c**) after SNL or sham surgery. $n = 5$ rats/time point/group. **$P < 0.01$ versus the corresponding sham group by two-tailed unpaired Student's *t*-test. Scale bar: 50 μm. (**d**) *Dnmt3a* mRNA expression in the ipsilateral L5 DRG on days 0, 3, and 7 after SNL or sham surgery. $n = 6$ rats/time point/group. Two-way ANOVA followed by *post hoc* Tukey test, $F_{time}$ (2, 48) = 7.4. *$P < 0.05$ versus the corresponding control group (0 day), (**e,f**) DNMT3a and DNMT3b proteins (**e**) and their mRNAs (**f**) in the ipsilateral L4 and L5 DRG on day 7 after CCI or sham surgery. $n = 6$ rats/group. *$P < 0.05$ or **$P < 0.01$ versus the corresponding sham group by two-tailed unpaired Student's *t*-test. Full-length blots are presented in Supplementary Fig. 6.

To further examine whether OCT1 directly regulates *Dnmt3a* transcription, we first carried out luciferase assay on transfected human embryonic kidney (HEK)-293T cells. Co-transfection of *Oct1* vector (encoding full-length OCT1), but not control *GFP* vector, significantly increased the activity of the *Dnmt3a* gene promoter including the OCT1-binding motif (Fig. 3c).

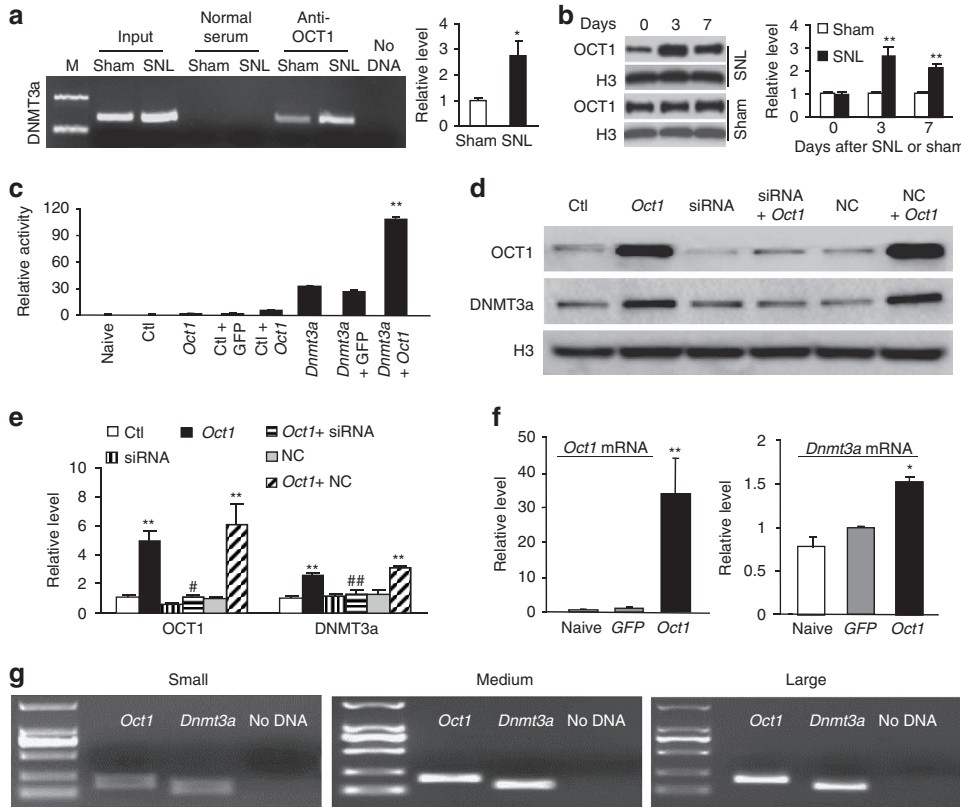

**Figure 3 | OCT1-triggered *Dnmt3a* gene transcription following peripheral nerve injury in rat injured DRG.** (**a**) The *Dnmt3a* promotor fragment immunoprecipitated by rabbit anti-OCT1 antibody in the ipsilateral L5 DRG on day 7 post-SNL or sham surgery. Input, total purified fragments. M, ladder marker. n = 3 repeats. *$P < 0.05$ versus the sham group by two-tailed unpaired Student's *t*-test. (**b**) OCT1 expression in the ipsilateral L5 DRG after SNL or sham surgery. n = 6 rats/time point/group. Two-way ANOVA followed by *post hoc* Tukey test, $F_{time}$ (2, 17) = 33.0. **$P < 0.01$ versus the corresponding control group (0 day). Full-length blots are presented in Supplementary Fig. 6. (**c**) *Dnmt3a* gene promotor activity in HEK-293 T cells transfected with the vectors and/or siRNAs as shown. Ctl: control empty pGL3-Basic. n = 3 repeats per treatment. One-way ANOVA followed by *post hoc* Tukey test, $F_{group}$ (7, 23) = 4,501.1. **$P < 0.01$ versus the pGL3-*Dnmt3a* vector (*Dnmt3a)* alone. (**d**,**e**) Levels of OCT1 and DNMT3a proteins in PC12 cells transfected with the vectors as shown. Ctl: control pHpa-trs-GFP. *Oct1*: pHpa-trs-*Oct1*. siRNA: *Oct1* siRNA. NC: negative control siRNA. N = 5 repeats per treatment. One-way ANOVA followed by *post hoc* Tukey test, $F_{group}$ (5, 17) = 13.8 for OCT1 and $F_{group}$ (5, 17) = 20.9 for DNMT3a. **$P < 0.01$ versus control GFP-treated group. #$P < 0.05$, ##$P < 0.01$ versus the *Oct1*-treated group. Full-length blots are presented in Supplementary Fig. 6. (**f**) Levels of *Oct1* mRNA and *Dnmt3a* mRNA in rat lumbar DRG cultured neurons transduced with AAV5-*GFP* (*GFP*) or AAV5-*Oct1* (*Oct1*). n = 3 repeats per treatment. One-way ANOVA followed by *post hoc* Tukey test, $F_{group}$ (2, 8) = 14.7 for *Oct1* mRNA and $F_{group}$ (2, 8) = 10.2 for *Dnmt3a* mRNA. *$P < 0.05$ or **$P < 0.01$ versus the corresponding naive condition. (**g**) Co-expression of *Oct1* mRNA with *Dnmt3a* mRNA in individual small, medium, and large neurons from the rat lumbar DRG. n = 3 repeats.

Overexpression of full-length OCT1 also markedly increased endogenous DNMT3a protein in *in vitro* cultured rat PC12 cells (Fig. 3d,e). This increase was abolished in the cells co-transfected with OCT1-specfic short interfering RNA (siRNA), but not negative control siRNA, (Fig. 3d,e), indicating that the *Dnmt3a* increase was a specific response to OCT1. We further confirmed the OCT1-triggered increase of *Dnmt3a* mRNA in cultured DRG neurons that were transduced with recombinant adeno-associated virus 5 (AAV5) that expressed full-length OCT1 (Fig. 3f). Single-cell RT-PCR analysis revealed co-expression of *Oct1* mRNA with *Dnmt3a* mRNA in individual small, medium or large DRG neurons (Fig. 3g). Taken together, our findings suggest that OCT1 participates in the nerve injury-induced upregulation of DRG *Dnmt3a*.

**Blocking increased DRG DNMT3a attenuates neuropathic pain.**
Is the increased DNMT3a in the injured DRG at the early time points involved in neuropathic pain genesis? To address this question, we examined the effect of DNMT3a knockdown in the injured DRG on the induction of SNL-induced pain

hypersensitivity. Specificity and selectivity of *Dnmt3a* shRNA were demonstrated by its ability to knockdown DNMT3a, but not DNMT1 and DNMT3b, in *in vitro* transfected HEK-293T cells (Supplementary Fig. 2a). Microinjection of AAV5 that expresses *Dnmt3a* shRNA (AAV5-*Dnmt3a* shRNA), but not AAV5 that expresses control scrambled shRNA (AAV5-scambled shRNA), into the ipsilateral L5 DRG 5 weeks before SNL blocked SNL-induced increases in *Dnmt3a* mRNA and protein on day 7 post-SNL (Fig. 4a,b). AAV5-*Dnmt3a* shRNA also reduced basal expression of *Dnmt3a* mRNA, but not protein, in the ipsilateral L5 DRG of sham rats (Fig. 4a,b). Microinjection of AAV5-*Dnmt3a* shRNA, but not control AAV5-scrambled shRNA, attenuated SNL-induced decreases in paw withdrawal thresholds to mechanical stimulation and paw withdrawal latencies to thermal or cold stimulation on the ipsilateral side from day 3 to 7 post-SNL (Fig. 4c–e). No changes were observed in basal mechanical, thermal, or cold responses on the contralateral side of SNL rats and on both ipsilateral and contralateral sides of sham rats following DRG microinjection of either viral shRNA (Fig. 4c–e; Supplementary Fig. 2b,c). Similar behavioural responses were seen after microinjection of

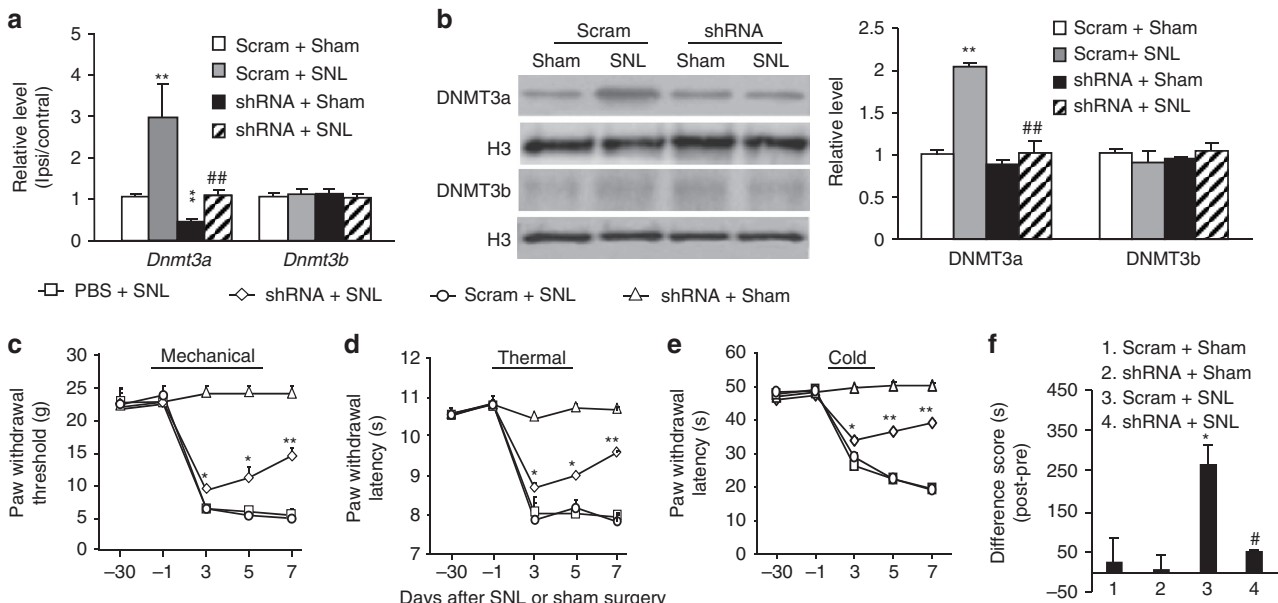

**Figure 4 | Blocking DRG DNMT3a increase attenuates neuropathic pain development in rats.** Scram: AAV5-scrambled *Dnmt3a* shRNA. shRNA: AAV5-*Dnmt3a* shRNA. (**a**) Levels of *Dnmt3a* and *Dnmt3b* mRNAs in the ipsilateral and contralateral L5 DRG on day 7 post-SNL or sham surgery from the treated groups as shown. $n = 6$ rats per group. One-way ANOVA followed by *post hoc* Tukey test, $F_{group}$ (3, 15) = 7.25 for *Dnmt3a* mRNA and $F_{group}$ (3, 15) = 0.29 for *Dnmt3b* mRNA. **$P < 0.01$ versus the corresponding scram plus sham group. (**b**) Levels of DNMT3a and DNMT3b proteins in the ipsilateral L5 DRG on day 7 post-SNL or sham surgery from the treated groups as shown. $n = 6$ rats per group. One-way ANOVA followed by *post hoc* Tukey test, $F_{group}$ (3, 11) = 46.6 for DNMT3a protein and $F_{group}$ (3, 11) = 0.54 for DNMT3b protein. **$P < 0.01$ versus the corresponding scram plus sham group. ##$P < 0.01$ versus the corresponding scram plus SNL group. Full-length blots are presented in Supplementary Fig. 6. (**c–e**) The effect of microinjection of AAV5-*Dnmt3a* shRNA, AAV5-scrambled *Dnmt3a* shRNA, or PBS into the ipsilateral L5 DRG on paw withdrawal responses to mechanical (**c**), thermal (**d**) and cold (**e**) stimuli on the ipsilateral side at days shown before or after SNL or sham surgery in rats. $n = 5$ rats per group. Two-way ANOVA followed by *post hoc* Tukey test, $F_{group}$ (3, 124) = 52.2 for (**c**), $F_{group}$ (3, 124) = 81.3 for (**d**) and $F_{group}$ (3, 124) = 611.1 for (**e**). *$P < 0.05$ or **$P < 0.01$ versus the corresponding PBS plus SNL group. (**f**) The effect of microinjection of AAV5-*Dnmt3a* shRNA or AAV5-scrambled *Dnmt3a* shRNA into the ipsilateral L5 DRG on the duration of time spent in saline- or lidocaine-paired chambers on day 7 post-SNL or sham surgery. Difference scores = post-conditioning time (post)—pre-conditioning time (pre) spent in the lidocaine-paired chamber. $n = 5$ rats/group. One-way ANOVA followed by *post hoc* Tukey test, $F_{group}$ (3, 19) = 6.3. *$P < 0.01$ versus the scram plus sham group. #$P < 0.01$ versus the scram plus SNL group.

AAV5-*Dnmt3a* shRNA or AAV5-scrambled shRNA in the CCI model (Supplementary Fig. 2d–h). In addition to SNL-induced evoked pain hypersensitivity, SNL produced stimulation-independent spontaneous ongoing pain indicated by obvious preference (that is, spend more time) for the lidocaine-paired chamber in the SNL rats microinjected with the AAV5-scrambled shRNA (Fig. 4f; Supplementary Fig. 2i). In contrast, SNL rats from the AAV5-*Dnmt3a* shRNA plus SNL group did not display marked preference for either the saline- or lidocaine-paired chamber, demonstrating no significant spontaneous pain (Fig. 4f; Supplementary Fig. 2i). As expected, sham rats microinjected with either viral shRNA did not show any preference for the saline- or lidocaine-paired chamber (Fig. 4f; Supplementary Fig. 2i).

Given that shRNA may have off-target effects, we further confirmed the role of DRG DNMT3a in neuropathic pain using microinjection of AAV5-*Cre* into the ipsilateral L4 DRG of *Dnmt3a*^fl/fl mice 30 days before unilateral L4 SNL (ref. 38). AAV5-*GFP* was used as a control. Specific and selective blockage of SNL-induced increases in the amounts of *Dnmt3a* mRNA and protein were seen in the injured DRG 7 days after SNL in the *Dnmt3a*^fl/fl mice injected with AAV5-*Cre* (Fig. 5a,b). Although AAV5-*Cre* injection reduced the basal level of *Dnmt3a* mRNA in the sham group (Fig. 5a,b), basal paw withdrawal responses to mechanical, thermal, and cold stimuli were similar between the two virus-injected groups (Fig. 5c–e). SNL-induced pain hypersensitivities were ameliorated on the ipsilateral side of the

AAV5-*Cre*-injected group (Fig. 5c–e). Compared to the AAV5-*GFP*-injected mice, paw withdrawal frequency to mechanical stimulation was lower and paw withdrawal latencies to thermal or cold stimulation were higher from days 3 to 7 post-SNL (Fig. 5c–e).

Given that neither shRNA-injected rats nor *Cre*-injected *Dnmt3a*^fl/fl mice displayed changes in locomotor function (Supplementary Table 1), our results strongly indicate that early increased DRG DNMT3a is required for neuropathic pain development.

**DRG DNMT3a overexpression leads to pain hypersensitivity.** We further asked whether the early increase in DRG DNMT3a was sufficient for neuropathic pain induction. To this end, we microinjected AAV5 that expressed full-length *Dnmt3a* (AAV5-*Dnmt3a*) into unilateral L4 and L5 DRG of naive adult rats. AAV5-*GFP* was used as a control. As expected, substantial increases in the amounts of *Dnmt3a* (but not *Dnmt1* and *Dnmt3b*) mRNA and protein were detected 4 weeks post-injection of AAV5-*Dnmt3a*, but not AAV5-*GFP*, in the injected DRG (Fig. 6a,b). The rats injected with AAV5-*Dnmt3a* (not AAV5-*GFP*) exhibited significant decreases in paw withdrawal thresholds in response to mechanical stimulation and in paw withdrawal latencies in response to thermal and cold stimuli on the ipsilateral side (Fig. 6c–e). These decreases occurred at 4 weeks post-injection and persisted for at

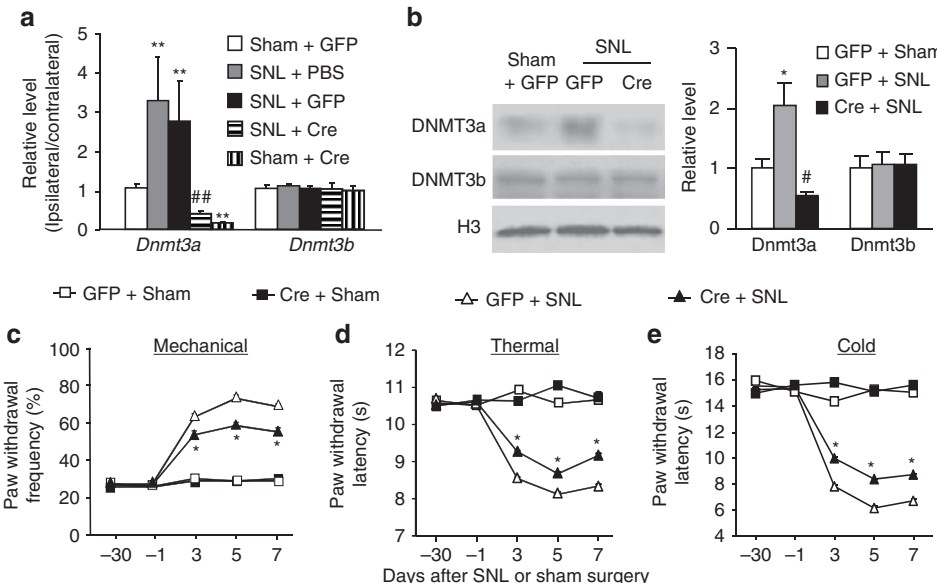

**Figure 5 | DRG DNMT3a knockdown attenuates neuropathic pain development in mice.** Cre: AAV5-Cre. GFP: AAV5-GFP. (**a**) Levels of *Dnmt3a* and *Dnmt3b* mRNAs in the ipsilateral L4 DRG on day 7 post-SNL or sham surgery from the *Dnmt3a*$^{fl/fl}$ mice injected with PBS, AAV5-*Cre* or AAV5-*GFP*. $n = 12$ mice per group. One-way ANOVA followed by *post hoc* Tukey test, $F_{group} (4, 16) = 5.45$ for *Dnmt3a* mRNA and $F_{group} (4, 16) = 0.15$ for *Dnmt3b* mRNA. **$P < 0.01$ versus the corresponding GFP plus sham group. $^{##}P < 0.01$ versus the corresponding PBS plus SNL group. (**b**) Levels of DNMT3a and DNMT3b proteins in the ipsilateral L4 DRG on day 7 post-SNL or sham surgery from the *Dnmt3a*$^{fl/fl}$ mice injected with AAV5-*Cre* or AAV5-*GFP*. $n = 12$ mice per group. One-way ANOVA followed by *post hoc* Tukey test, $F_{group} (2, 8) = 10.5$ for DNMT3a protein and $F_{group} (2, 8) = 0.05$ for DNMT3b protein. *$P < 0.05$ versus the corresponding GFP plus sham group. #$P < 0.05$ versus the corresponding GFP plus SNL group. Full-length blots are presented in Supplementary Fig. 6. (**c–e**) The effect of microinjection of AAV5-*Cre* or AAV5-*GFP* into the ipsilateral L4 DRG of *Dnmt3a*$^{fl/fl}$ mice on paw withdrawal responses to mechanical (**c**), thermal (**d**) and cold (**e**) stimuli on the ipsilateral side at days shown before or after SNL or sham surgery. $n = 8$ mice per group. Two-way ANOVA followed by *post hoc* Tukey test, $F_{group} (3, 129) = 238.9$ for (**c**), $F_{group} (3, 129) = 357.9$ for (**d**) and $F_{group} (3, 129) = 367.9$ for (**e**). *$P < 0.05$ versus the corresponding GFP plus SNL group.

least 8 weeks (Fig. 6c–e). Viral injection did not affect locomotor functions (Supplementary Table 1) or alter the basal paw withdrawal responses on the contralateral side (Fig. 6c,d). Similar results were observed in naive mice with microinjection of herpes simplex virus (HSV) that expresses full-length *Dnmt3a* into unilateral L3/4 DRG (Fig. 6f–i; Supplementary Fig. 3a–c). The findings indicate that the increased DNMT3a in DRG leads to mechanical allodynia and thermal and cold hyperalgesia, major clinical symptoms of neuropathic pain, in the absence of nerve injury. These behavioural observations were further supported by the following evidence of spinal cord dorsal horn central sensitization. The levels of phosphorylated extracellular signal-regulated kinase 1/2 (p-ERK1/2, a marker for neuronal hyperactivation) and glial fibrillary acidic protein (GFAP, a marker for astrocyte hyperactivation) significantly increased in the ipsilateral L3/4 dorsal horn of spinal cord on day 6 after microinjection of HSV-*Dnmt3a* compared to those after microinjection of HSV-*GFP* (Fig. 6j).

**DNMT3a represses *Kcna2* in DRG neurons.** Next, we explored the mechanism by which DNMT3a in DRG contributes to neuropathic pain development. Nerve injury-induced downregulation of DRG *Kcna2* is an endogenous instigator of neuropathic pain genesis[7–9]. We found that microinjection of AAV5-*Dnmt3a* shRNA, but not AAV5-scrambled shRNA, significantly reversed the decrease of *Kcna2* mRNA and protein in the injured L5 DRG on day 7 post-SNL (Fig. 7a,b). Similarly, microinjection of AAV5-*Cre*, but not AAV5-*GFP*, into the ipsilateral L4 DRG of *Dnmt3a*$^{fl/fl}$ mice markedly restored the expression of *Kcna2* mRNA and protein in the injured

L4 DRG on day 7 post-SNL (Fig. 7c,d). Neither AAV5-*Dnmt3a* shRNA nor AAV5-*Cre* affected SNL-induced reductions of *Kcna1* and *Kcna4* mRNAs in the injured DRG (Fig. 7a–d). Furthermore, microinjection of AAV5-*Dnmt3a*, but not AAV5-*GFP*, into the unilateral L4 and L5 DRG dramatically decreased the levels of *Kcna2* mRNA and protein in the injected DRG 5 weeks post-injection (Fig. 8a,b). Basal expression of mRNAs and proteins of *Kcna1* and *Kcna4* did not change in the injected DRGs (Fig. 8a,b). Given that approximately 62.1% of Kcna2-labelled DRG neurons were positive for DNMT3a (Fig. 8c), it is very likely that DNMT3a directly regulates DRG *Kcna2* expression in DRG. Indeed, overexpression of DNMT3a through transduction of HSV-*Dnmt3a* in cultured DRG neurons of *Dnmt3a*$^{fl/fl}$ mice reduced the level of *Kcna2* mRNA, whereas knockdown of DNMT3a through transduction of AAV5-*Cre* in *in vitro* cultured DRG neurons of *Dnmt3a*$^{fl/fl}$ mice substantially increased Kcna2 mRNA expression (Fig. 8d). This increase could be blocked by co-transduction of AAV5-*Cre* and HSV-*Dnmt3a* (Fig. 8d). The evidence described above strongly suggests the participation of DNMT3a in nerve injury-induced *Kcna2* downregulation in the injured DRG.

We further used ChIP assays and found that DNMT3a binds to two regions ($-663/-389$ bp and $-491/-199$ bp) of the *Kcna2* gene promoter, as demonstrated by the amplification of only these two regions (out of 7 regions from $-663$ to $+928$ bp) from the complexes immunoprecipitated with DNMT3a antibody in nuclear fractions from sham DRG (Fig. 9a). The binding activities in these two regions in the injured DRG on day 7 after SNL increased by 3.9-fold and 4.4-fold, respectively, compared to those after sham surgery (Fig. 9b). Furthermore, these increased binding activities affected the DNA methylation pattern

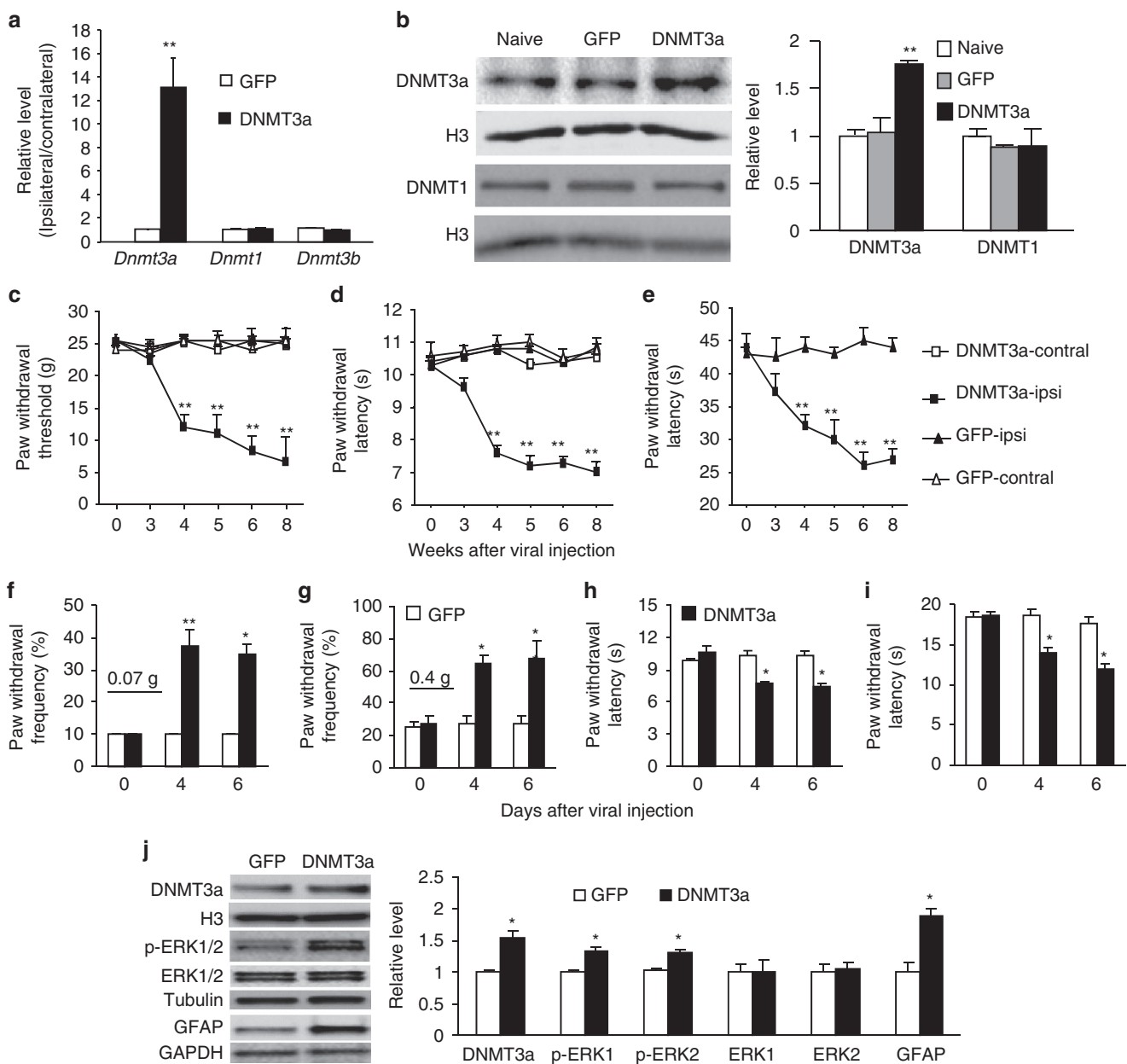

**Figure 6 | DRG *Dnmt3a* overexpression produces neuropathic pain symptoms in both rats and mice.** GFP: AAV5-*GFP* in **a–e** or HSV-*GFP* in **f–j**. DNMT3a: AAV5-*Dnmt3a* in **a–e** or HSV-*Dnmt3a* in **f–j**. (**a**) Amounts of *Dnmt3a, Dnmt1,* and *Dnmt3b* mRNAs in the ipsilateral and contralateral L4/5 DRG 8 weeks after viral microinjection into the ipsilateral L4/5 DRG in rats. $n = 3$ rats per group. **$P < 0.01$ versus the corresponding GFP group by two-tailed unpaired Student's *t*-test. (**b**) Levels of DNMT3a and DNMT1 proteins in the ipsilateral L4/5 DRG in naive rats and 8 weeks after viral microinjection into the ipsilateral L4/5 DRG in rats. $n = 3$ rats/group. One-way ANOVA followed by *post hoc* Tukey test, $F_{group}$ (2, 8) = 40.7 for DNMT3a and $F_{group}$ (2, 8) = 1.21 for DNMT1. **$P < 0.01$ versus naive rats. Full-length blots are presented in Supplementary Fig. 6. (**c–e**) Paw withdrawal responses to mechanical (**c**), thermal (**d**) and cold (**e**) stimuli on the ipsilateral (ipsi) and contralateral (contral) sides at time points as shown after viral microinjection into the ipsilateral L4/5 DRG in rats. $n = 10$ rats per group. Two-way ANOVA followed by *post hoc* Tukey test, $F_{group}$ (3, 119) = 206.9 for (**c**), $F_{group}$ (3, 119) = 69.1 for (**d**), and $F_{group}$ (1, 59) = 178.0 for (**e**). **$P < 0.01$ versus the corresponding GFP group on the ipsilateral side. (**f–i**) Ipsilateral paw withdrawal responses to mechanical (**f,g**), thermal (**h**) and cold (**i**) stimuli at time points as shown after viral microinjection into the ipsilateral L3/4 DRG in mice. $n = 10$ mice per group. Two-way ANOVA followed by *post hoc* Tukey test, $F_{group}$ (1, 23) = 88.2 for (**f**), $F_{group}$ (1, 23) = 28.4 for (**g**), $F_{group}$ (1, 23) = 25.2 for (**h**) and $F_{group}$ (1, 23) = 37.8 for (**i**). *$P < 0.05$ or **$P < 0.01$ versus the corresponding GFP group. (**j**) Levels of DNMT3a, p-ERK1/2, ERK1/2 and GFAP in the ipsilateral L3/4 dorsal horn 6 days after viral microinjection into the ipsilateral L3/4 DRG in mice. $n = 12$ mice per group. *$P < 0.05$ versus the corresponding HSV-*GFP* group by two-tailed unpaired Student's *t*-test. Full-length blots are presented in Supplementary Fig. 6.

and level in the promoter region of the *Kcna2* gene after SNL. The bisulfite clone-sequencing assay showed increases in DNA methylation at $-457$, $-444$, $-440$, and $-374$ CpG sites from $-540$ to $+500$ bp of the *Kcna2* gene consisting of 65 CpG sites (Fig. 9c). The bisulfite pyro-sequencing assay further

confirmed the increases in DNA methylation at $-457$ and $-444$ CpG sites, although an increase was detected at an additional -482 CpG site (Fig. 9d). The $-457$ and $-444$ CpG sites are located within DNMT3a binding regions described above. The increased methylations at these two sites may be dependent on

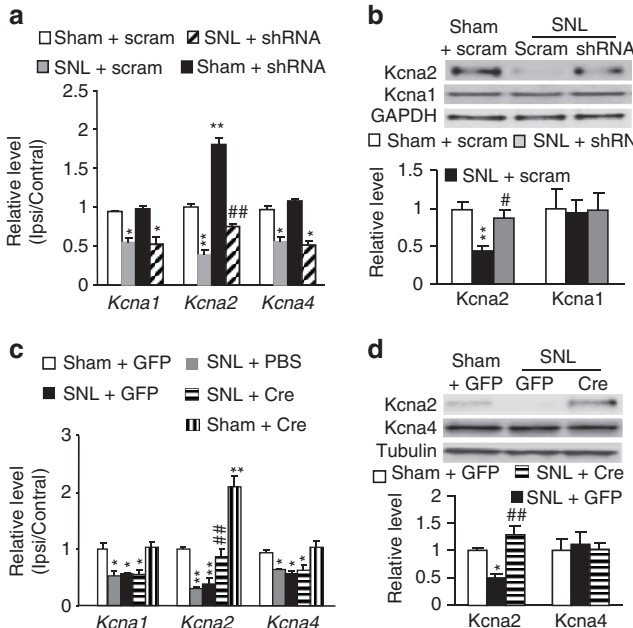

**Figure 7 | Blocking DRG DNMT3a increase rescues Kcna2 expression in the injured DRG of rats or mice post-SNL.** (a,b) Levels of *Kcna1*, *Kcna2*, and *Kcna4* mRNAs in the ipsilateral (Ipsi) or contralateral (Contral) L5 DRG (**a**) and levels of Kcna1 and Kcna2 proteins in the ipsilateral L5 DRG (**b**) from the AAV5-*Dnmt3a* shRNA (shRNA)- or AAV5-*Dnmt3a* scrambled shRNA (scram)-injected rats on day 7 post-SNL or sham surgery. n = 6 rats/group. One-way ANOVA followed by *post hoc* Tukey test, $F_{group} (3, 15) = 18.9$ for *Kcna1* mRNA, $F_{group} (3, 15) = 128.6$ for *Kcna2* mRNA, $F_{group} (3, 15) = 38.9$ for *Kcna4* mRNA, $F_{group} (2, 8) = 0.02$ for Kcna1 protein, and $F_{group} (2, 8) = 12.2$ for Kcna2 protein. *$P < 0.05$ or **$P < 0.01$ versus the corresponding sham plus scram group. #$P < 0.05$ or ##$P < 0.01$ versus the corresponding SNL plus scram group. Full-length blots are presented in Supplementary Fig. 6. (**c,d**) Levels of *Kcna1*, *Kcna2* and *Kcna4* mRNAs in the ipsilateral (Ipsi) or contralateral (Contral) L4 DRG (**c**) and levels of Kcna2 and Kcna4 proteins in the ipsilateral L4 DRG (**d**) from the AAV5-*Cre* (Cre)- or AAV5-*GFP* (GFP)-injected *Dnmt3a*<sup>fl/fl</sup> mice on day 7 post-SNL or sham surgery. n = 12 mice/group. One-way ANOVA followed by *post hoc* Tukey test, $F_{group} (4, 16) = 9.68$ for *Kcna1* mRNA, $F_{group} (4, 16) = 36.9$ for *Kcna2* mRNA, $F_{group} (4, 16) = 6.40$ for *Kcna4* mRNA, $F_{group} (2, 8) = 13.4$ for Kcna2 protein, and $F_{group} (2, 8) = 0.10$ for Kcna4 protein. *$P < 0.05$ or **$P < 0.01$ versus the corresponding sham plus GFP group. ##$P < 0.01$ versus the corresponding SNL plus GFP group. Full-length blots are presented in Supplementary Fig. 6.

DNMT3a, as DRG DNMT3a knockdown abolished the SNL-induced increases in the methylation levels at the $-457$ and $-444$ sites (Fig. 9e). Furthermore, these two sites may be relevant to *Kcna2* transcription because the *Kcna2* promoter activity increased significantly upon the deletion of the $-457$ or $-444$ CpG site (Supplementary Fig. 4a). DNMT3a overexpression markedly reduced *Kcna2* promoter activity, which could be blocked when both $-457$ and $-444$ CpG sites were deleted (Supplementary Fig. 4b). These findings indicate that DNMT3a-triggered DNA methylation at $-457$ and $-444$ sites of the *Kcna2* promoter region may be involved in nerve injury-induced silencing of the *Kcna2* gene in injured DRG.

**DRG DNMT3a overexpression increases neuronal excitability.** Finally, we investigated whether mimicking the SNL-induced DRG DNMT3a increase would affect total Kv current in DRG neurons. Whole-cell voltage-clamp recording was carried out 5–8 weeks after microinjection of AAV5-*GFP* alone (control group) or a mixed viral solution of AAV5-*Dnmt3a* plus AAV5-*GFP* (DNMT3a-injected group) into the DRG. Only green DRG neurons were recorded. Compared to the control group, total Kv current density was significantly decreased in large, medium, and small DRG neurons from the DNMT3a-injected group (Fig. 10a,b; Supplementary Fig. 5a–f). To demonstrate whether *Kcna2* downregulation contributed to this reduction, we carried out bath application of 100 nM maurotoxin (MTX), a selective Kcna2 current inhibitor[8,9,39,40]. After MTX treatment, the decreases in total Kv currents in large and medium neurons from the control group were greater than those from the DNMT3a-injected group (Fig. 10a,b; Supplementary Fig. 5a–f). When tested at $+50$ mV, total Kv currents in large and medium neurons from the control group were reduced by 28% and 32%, respectively, compared to those before MTX treatment, whereas total Kv currents in large and medium neurons from the DNMT3a-injected group were reduced only by 13% compared to those before MTX treatment (Fig. 10c; Supplementary Fig. 5c). In small DRG neurons, MTX did not produce marked current reductions or significant differences between control and DNMT3a-injected groups (Supplementary Fig. 5d–f). The evidence indicates that only large and medium DRG neurons display reductions in Kcna2-related current although all DRG neurons exhibit decreases in total Kv current densities after DRG DNMT3a overexpression.

DRG neuronal excitability was also examined. Whole-cell current-clamp recording was carried out 5–8 weeks after viral injection. Compared to the control group, large, medium, and small neurons in the DNMT3a-injected group showed increases by 11.98, 12.01 and 7.63 mV, respectively, in the resting membrane potentials (Fig. 10d) and decreases by 40%, 49% and 40%, respectively, in the current thresholds (Fig. 10e). Moreover, DNMT3a injection significantly increased the numbers of action potentials evoked by stimulation of $\geq 200$ pA in large, medium, and small neurons (Fig. 10f–i), although this injection did not alter the membrane input resistances or other action potential parameters, such as amplitude, threshold, duration, overshoot and afterhyperpolarization amplitude (Supplementary Table 2). These data indicate a significant increase in DRG neuronal excitability after DRG DNMT3a overexpression.

## Discussion

In this study, we demonstrated that peripheral nerve injury led to an increase in DNMT3a expression through an activation of the transcription factor OCT1 in the injured DRG neurons. This increase correlates with an elevation in the level of DNA methylation at some CpG sites within the *Kcna2* promoter region and is associated with nerve injury-induced downregulation of *Kcna2* expression in the injured DRG. Given that DRG Kcna2 downregulation contributes to neuropathic pain genesis[7,9,27], DNMT3a likely participates in the mechanisms that underlie neuropathic pain development.

DNMT3a is expressed widely in most tissues including the nervous system[19]. A previous study reported that DNMT3a is located in DRG satellite cells[41]. However, this conclusion remains uncertain because no specific neuronal and glial markers were used[41]. Moreover, the specificity and selectivity of the antibody used were not addressed[41]. The present study showed that DNMT3a was co-expressed with NeuN in neuronal nuclei and was not detected in GS-positive satellite cells. DNMT3a co-localized with CGRP, IB4 or NF200, which are distributed

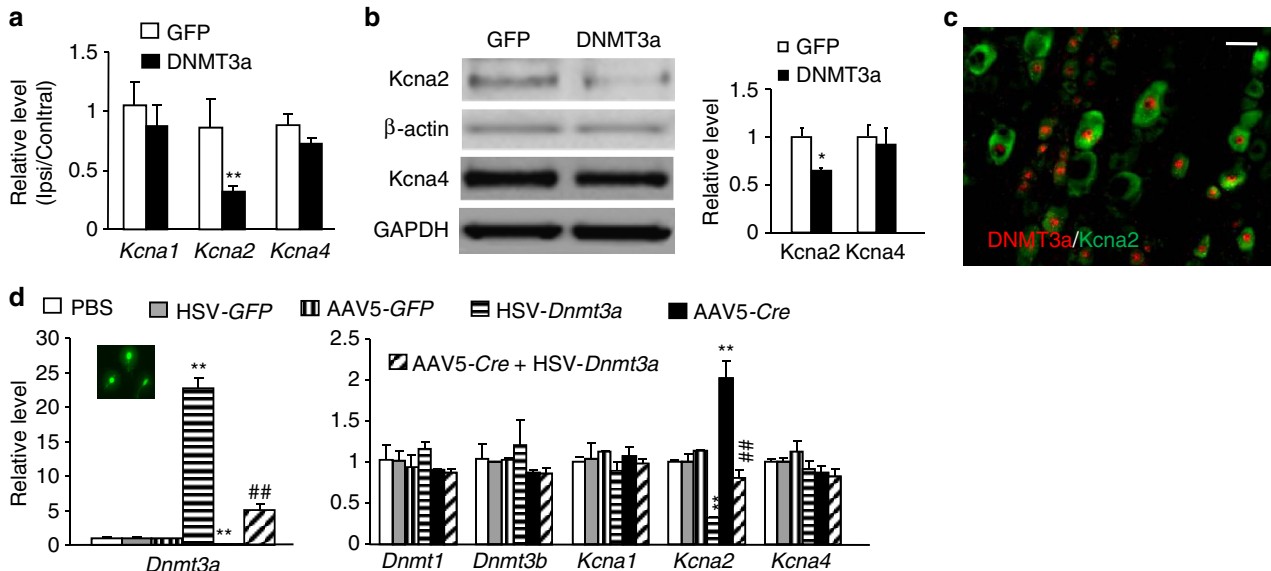

**Figure 8 | DRG DNMT3a overexpression represses *Kcna2* expression.** (**a,b**) Levels of *Kcna1*, *Kcna2*, and *Kcna4* mRNAs on the ipsilateral (Ipsi) or contralateral (Contral) sides (**a**) and levels of Kcna2 and Kcna4 proteins on the ipsilateral side (**b**) 8 weeks after microinjection of AAV5-*Dnmt3a* (DNMT3a) or AAV5-*GFP* (GFP) into the unilateral L4/5 DRG. n = 6 rats/group. *$P < 0.05$ or **$P < 0.01$ versus the corresponding GFP group by two-tailed unpaired Student's *t*-test. Full-length blots are presented in Supplementary Fig. 6. (**c**) Co-localization of DNMT3a with Kcna2 in rat L5 DRG neurons. Scale bar, 40 μm. (**d**) Amounts of *Dnmt3a, Dnmt1, Dnmt3b, Kcna1, Kcna2* and *Kcna4* mRNAs in *Dnmt3a*[fl/fl] mouse DRG cultured neurons transduced with PBS or virus as indicated. Inset: GFP-labelled neurons. n = 3 repeats per group. One-way ANOVA (relative level versus. group) followed by *post hoc* Tukey test, $F_{group} (5, 17) = 136.9$ for *Dnmt3a*, $F_{group} (5, 17) = 0.85$ for *Dnmt1*, $F_{group} (5, 17) = 0.67$ for *Dnmt3b*, $F_{group} (5, 17) = 0.56$ for *Kcna1*, $F_{group} (5, 17) = 38.1$ for *Kcna2*, and $F_{group} (5, 17) = 1.52$ for *Kcna4*. **$P < 0.01$ versus the corresponding PBS group. ##$P < 0.01$ versus the corresponding AAV5-*Cre*-treated group.

exclusively in DRG neurons. In addition, no immunoreactivity was detected in the DRG when the DNMT3a antibody was omitted or replaced with normal IgG. Our observations are consistent with those found in previous reports that showed a neuronal location for DNMT3a in the central nervous system including spinal cord, frontal cortex and amygdala[42,43].

The *Dnmt3a* gene in DRG can be activated at the transcriptional level in specific response to peripheral nerve injury. *Dnmt3a* mRNA and its protein increased in the injured DRG, but not in intact (uninjured) DRG or bilateral spinal cord after SNL or CCI. Interestingly, CFA injection into a hind paw did not alter basal expression of DNMT3a protein in the DRG and spinal cord on either ipsilateral or contralateral side, although a previous study showed a small increase in *Dnmt3a* mRNA in the ipsilateral dorsal horn 7 days after CFA injection into the ankle joint[44]. *Dnmt3a* gene activation appears to be tissue- and nerve injury-specific. As expected, the SNL-induced increase in DRG DNMT3a could be blocked specifically and selectively via the injection of AAV5-*Dnmt3a* shRNA into rat DRG or the injection of AAV5-*Cre* into the *Dnmt3a*[fl/fl] mouse DRG. These viral injections also reduced basal *Dnmt3a* mRNA (but not protein) expression in the injected DRGs of the sham animals. The reason for no effect of viral injections on baseline DNMT3a protein is unknown, but it is possible that the remaining *Dnmt3a* mRNA after its knockdown may have a high translational efficacy, thus a normal level of basal DNMT3a protein is maintained in the injected DRG. We further demonstrated that SNL-induced *Dnmt3a* gene expression was regulated by the transcription factor OCT1 in the injured DRG. Whether additional transcription factors are also involved in *Dnmt3a* gene activation and whether OCT1 also regulates the expression of other genes relevant to nociceptive processing remains to be determined. Finally, other potential

possibilities such as increased mRNA stability and/or epigenetic modifications[9] that may lead to an increase in the level of *Dnmt3a* mRNA could not be excluded under neuropathic pain conditions.

The increased DNMT3a may participate in nerve injury-induced *Kcna2* downregulation in the injured DRG. Blocking the SNL-induced increase in DRG DNMT3a via its genetic knockout or knockdown restored the expression of *Kcna2*, but not *Kcna1* and *Kcna4*, in the injured DRG, whereas mimicking the SNL-induced increase in DRG DNMT3a via its genetic overexpression decreased the expression of *Kcna2*, but not *Kcna4*, in the DRG of naive rat. Our *in vitro* DRG neuronal culture studies further demonstrated that DNMT3a directly regulated the expression of *Kcna2*, but not *Kcna1* and *Kcna4*. How DNMT3a specifically targets *Kcna2* is unknown, but DNMT3a participation in nerve injury-induced DRG *Kcna2* silencing may be related to the increased binding of DNMT3a to the *Kcna2* gene promoter region ( − 663/ − 199) and associated with an elevation in DNA methylation levels within this region after SNL. Indeed, the levels of DNA methylation significantly increased at − 457 and − 444 CpG sites within the region (from − 540 to + 500 bp) of the *Kcna2* gene containing 65 CpG sites on day 7 post-SNL. These increases are DNMT3a-dependent as blocking the SNL-induced upregulation in DRG DNMT3a abolished the SNL-induced elevation in *Kcna2* DNA methylation level at these two CpG sites. Whether the SNL-induced increase in DNMT3a-triggered DNA methylation occurs at other CpG sites within the *Kcna2* gene regions that were not studied is unknown, but DNMT3a overexpression significantly reduced *Kcna2* promoter activity, which could be rescued when both − 457 and − 444 CpG sites were deleted. Given that SNL did not alter the basal expression of another *de novo* methyltransferase DNMT3b, or any TETs (that lead to

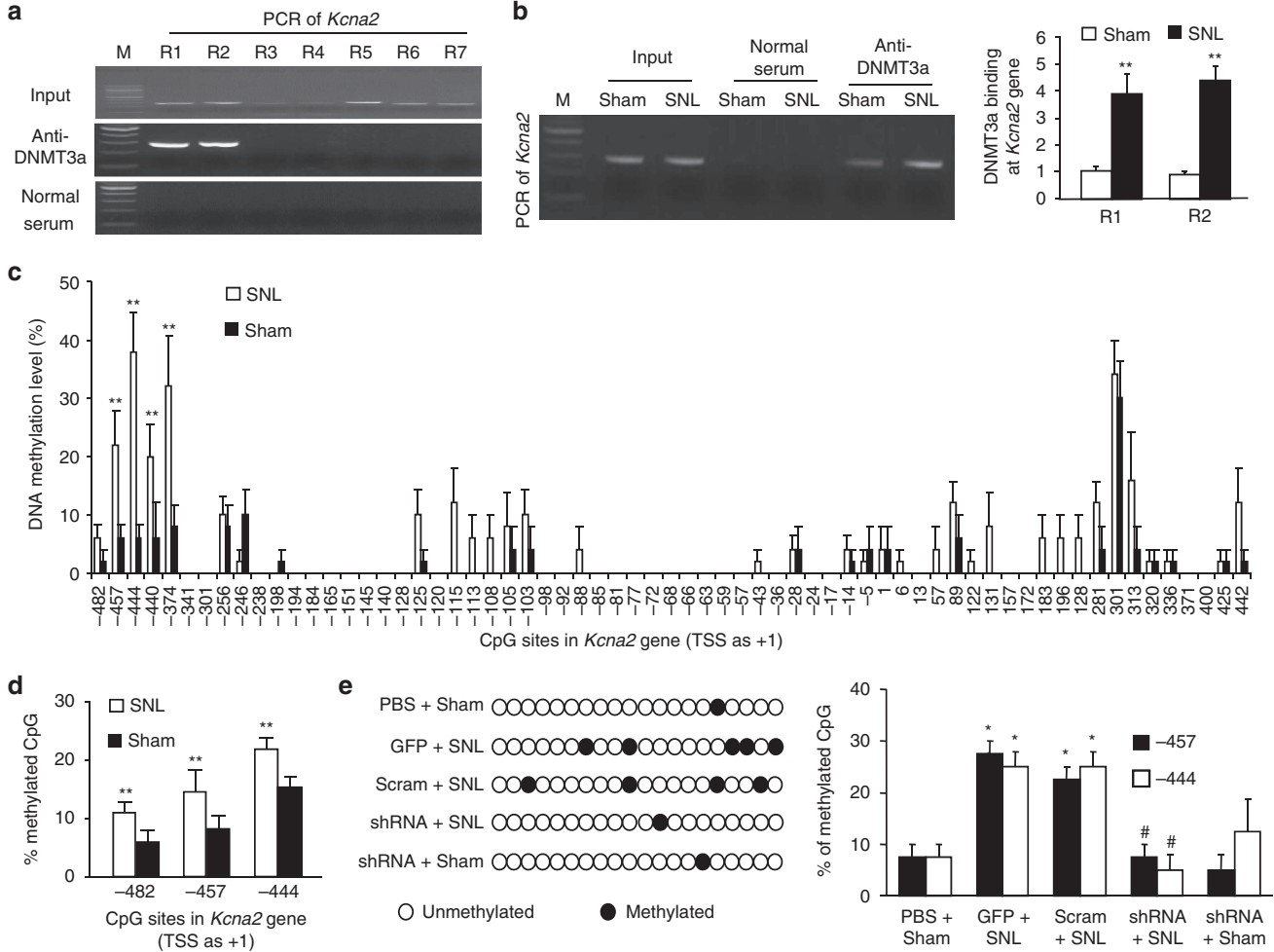

**Figure 9 | DNMT3a is required for the nerve injury-induced increase in *Kcna2* DNA methylation in rat injured DRG.** (**a**) Two regions (R1, −663/−389; R2, −491/−199), but not other regions (R3, −237/+42; R4, +20/+278; R5, +185/+464; R6: +411/+649; R7: +680/+928), from the *Kcna2* gene were immunoprecipitated by rabbit anti-DNMT3a (not by rabbit normal serum) in rat lumbar DRG. Input, total purified fragments. M, ladder marker. n = 3 repeats. (**b**) DNMT3a binding to R1 and R2 regions within the *Dnmt3a* gene in the ipsilateral L5 DRG of rats on day 7 post-SNL or sham surgery. Left: the binding of Dnmt3a to the R1 region of *Kcna2* gene. Right: the quantitative analysis of the binding. n = 3 repeats (3 rats/repeat)/group. **$P < 0.01$ versus the corresponding sham group by two-tailed unpaired Student's $t$-test. (**c,d**) The increases in the levels of DNA methylation at −457, −444, −440, and −374 CpG sites by bisulfite clone-sequencing assay (**c**) and at −482, −457, and −444 CpG sites by bisulfite pyro-sequencing assay (**d**) from −540 to 500 bp sites of *Kcna2* gene consisted of 65 CpG sites in the ipsilateral L5 DRG of rats on day 7 post-SNL. n = 3 repeats (6 rats/repeat)/group. **$P < 0.01$ versus the corresponding sham group by two-tailed unpaired Student's $t$-test. (**e**) The percentages of methylation at −457 and −444 CpG sites of the *Kcna2* gene in the ipsilateral L5 DRG on day 7 post-SNL or sham surgery from five groups as indicated. GFP: AAV5-*GFP*. scram: AAV5-*Dnmt3a* scrambled shRNA. shRNA: AAV5-*Dnmt3a* shRNA. Left: Representation of a single cloned allele per group at the -457 CpG site. Right: Statistical analysis at −457 and −444 CpG sites. n = 3 repeats (3 rats/repeat)/group. One-way ANOVA (methylation versus. group) followed by *post hoc* Tukey test, $F_{group}$ (3, 19) = 15.8 for −457 site and $F_{group}$ (3, 19) = 6.4 for −440 site. *$P < 0.05$ versus the corresponding PBS plus sham group. #$P < 0.05$ versus the corresponding GFP plus SNL group.

oxidation of methylated DNA and to demethylation) in the injured DRG, our findings suggest a specific contribution of DNMT3a-triggered DNA methylation at these two sites of the *Kcna2* promoter to nerve injury-induced *Kcna2* gene silencing in the injured DRG. SNL-induced elevation of DNA methylation in the *Kcna2* gene promoter may interfere with the binding of transcription factors and/or serve as docking sites for methyl-CpG-binding domain proteins to lead to its silence in the injured DRG (refs 45,46).

It should be noted that peripheral nerve injury leads to DRG Kcna2 downregulation through multiple mechanisms. Besides the role of DNMT3a shown in the present study and the function of endogenous long non-coding *Kcna2* antisense RNA revealed previously[9], we recently demonstrated that G9a-triggered histone methylation participated in nerve

injury-induced *Kcna2* silencing in the injured DRG (ref. 26). How these epigenetic mechanisms work together to regulate *Kcna2* expression and whether they interact with/affect each other under neuropathic pain conditions are unclear and will be addressed in future studies.

Nerve injury-induced abnormal ectopic firing and hyperexcitability occurring in DRG neurons are considered to play a key role in the genesis of neuropathic pain[3,4]. Nerve injury-induced silencing of *Kcna2* in the injured DRG produces hyperexcitability in the medium and large DRG neurons and contributes to neuropathic pain development[7,9,27]. Given that DNMT3a, like Kcna2, is predominantly expressed in the medium and large DRG neurons and participates in nerve injury-induced downregulation of *Kcna2* in the injured DRG, it is very likely that the nerve injury-induced increase of DNMT3a in the injured

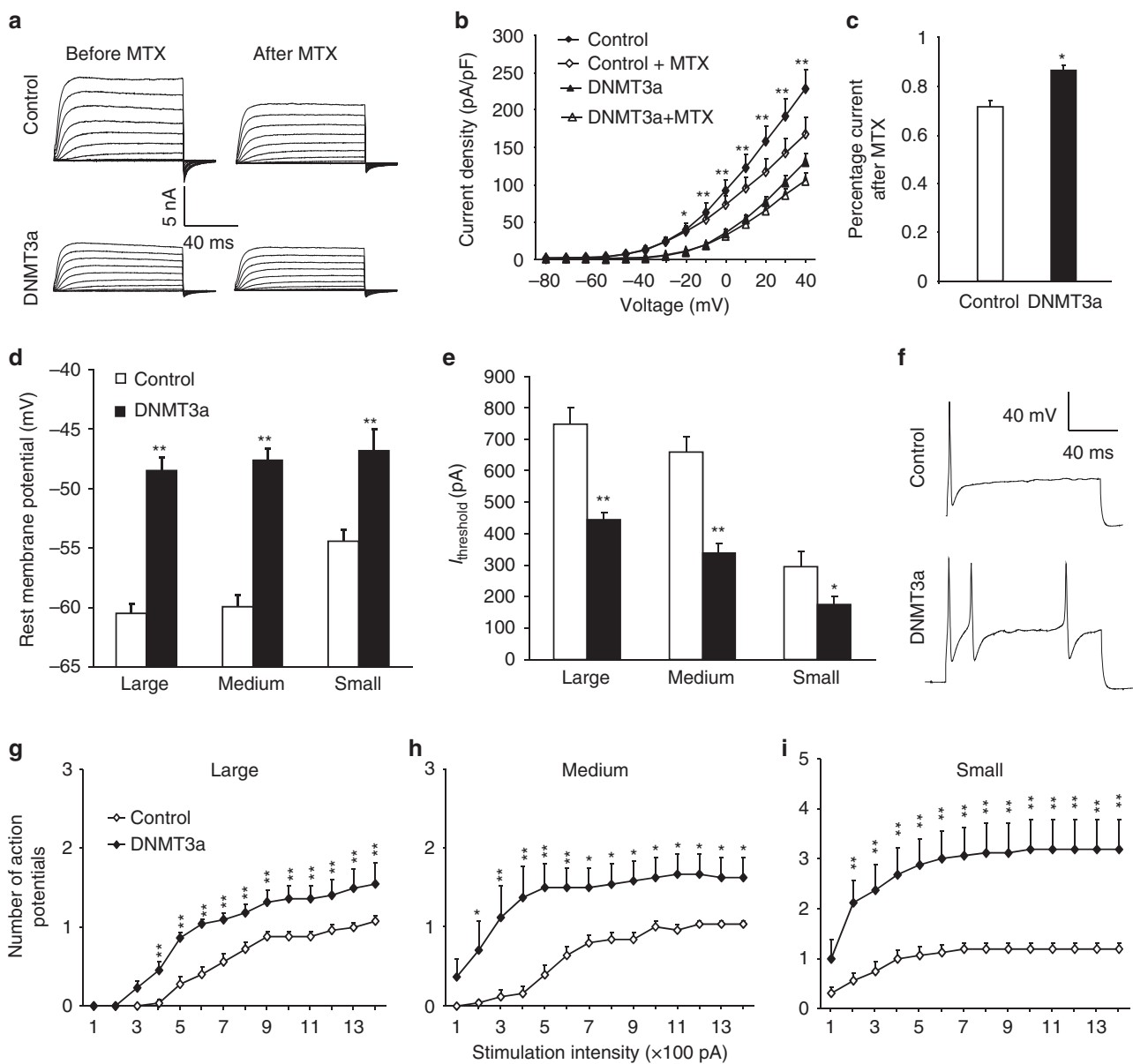

**Figure 10 | *Dnmt3a* overexpression reduces total Kv current and increases the excitability in the injected DRG neurons 5–8 weeks after microinjection of AAV5-*GFP* (Control) or AAV5-*Dnmt3a* (DNMT3a) into the unilateral L4 and L5 DRG of rats.** (**a**) Representative traces of total Kv currents before or after bath perfusion of 100 nM maurotoxin (MTX) in the large DRG neurons. (**b**) *I–V* curves before or after 100 nM MTX treatment in the large DRG neurons from control ($n = 17$ neurons from 7 rats) and DNMT3a-treated ($n = 25$ neurons from 9 rats) groups. One-way ANOVA (current density versus. group) followed by *post hoc* Tukey test, $F_{group}$ (1, 13) = 145.1. *$P < 0.05$, **$P < 0.01$ versus the DNMT3a-treated group. (**c**) At $+50$ mV, reduction in total Kv current in the large DRG neurons after MTX treatment was greater in the control group than in the DNMT3a-treated group. *$P < 0.05$ versus control group by two-tailed unpaired Student's *t*-test. (**d,e**) Resting membrane potential (**d**) and current threshold for pulses ($I_{threshold}$, **e**). $n = 25$ large, 25 medium and 16 small neurons from control group (8 rats). $n = 22$ large, 24 medium and 16 small neurons from the DNMT3a-treated group (10 rats). *$P < 0.05$, **$P < 0.01$ versus the corresponding control group by two-tailed unpaired Student's *t*-test. (**f**) Representative traces of evoked action potentials in DRG neurons. (**g–i**) Numbers of evoked action potentials from control and DNMT3a-treated groups after application of different currents as indicated. Numbers of the recorded cells are the same as in d. Two-way ANOVA (effect versus. group × stimulation interaction) followed by *post hoc* Tukey test, $F_{group}$ (1, 13) = 98.3 for large, $F_{group}$ (1, 13) = 91.4 for medium and $F_{group}$ (1, 13) = 134.9 for small. *$P < 0.05$, **$P < 0.01$ versus the same stimulation intensity in the control group.

DRG elevates DRG neuronal excitability. Indeed, mimicking the nerve injury-induced DNMT3a increase in the injured DRG not only reduced total Kv current, but also depolarized the resting membrane potential, decreased the current threshold for action potential generation and increased the number of action potentials in medium and large DRG neurons. Interestingly, these phenomena were also seen in small

DRG neurons although Kcna2-mediated current was not altered due to poor Kcna2 expression in small DRG neurons[7,9,27]. Kcna1 and Kcna4 are expressed highly in small DRG neurons[17], but their mRNA downregulation caused by nerve injury could not be rescued by DNMT3a knockdown in the injured DRG. Given that DRG DNMT3a overexpression decreased total Kv current and increased neuronal excitability in small DRG

neurons, DNMT3a is likely involved in nerve injury-induced silencing of other Kv channels expressed in small DRG neurons. In addition, the participation of DNMT3a in nerve injury-induced downregulation of non-Kv channel genes relevant to neuronal excitability in the injured DRG cannot be ruled out at present. It is well documented that increased DRG neuronal excitability drives the release of neurotransmitters (such as glutamate, substance P and CGRP) from the primary afferent terminals which leads to spinal central sensitization and pain hypersensitivities under neuropathic pain conditions[2,4,47]. We further demonstrated that mimicking the nerve injury-induced increase of DRG DNMT3a expression induces expression of p-ERK1/2 and GFAP (the markers for central sensitization) in dorsal horn. More importantly, blocking the nerve injury-induced increase in DRG DNMT3a attenuated the development of nerve injury-induced mechanical allodynia and thermal and cold hyperalgesia. DNMT3a contributes to neuropathic pain genesis likely by repressing at least *Kcna2* in the injured DRG.

In summary, our study reveals a DNMT3a-triggered epigenetic mechanism of *Kcna2* downregulation in the injured DRG after peripheral nerve injury. Given that DRG *Kcna2* down-regulation contributes to the development of neuropathic pain in rodents[7,9,27] and that we find here that blocking the SNL-induced DRG DNMT3a increase impaired this disorder without altering acute pain and motor functions, DNMT3a may be a target for neuropathic pain management. However, it is worth noting that DNMT3a is expressed in other tissues and may target other genes in addition to *Kcna2*. Thus, potential side effects caused by DNMT3a inhibitors should be explored carefully.

## Methods

**Animals.** Male Sprague-Dawley rats weighing 200–250 g (2–3-month old) were purchased from Charles River Laboratories (Wilington, MA). *Dnmt3a*$^{fl/fl}$ mice were provided by Dr. Eric J Nestler (Icahn School of Medicine at Mount Sinai, NY) and bred in our facility. Male *Dnmt3a*$^{fl/fl}$ mice weighing 25–30 g (2–3-month old) were used for the experiments. All animals were kept in the central housing facility at Rutgers New Jersey Medical School under a standard 12-h light/dark cycle. Water and food pellets were supplied *ad libitum*. The Animal Care and Use Committee at the Rutgers New Jersey Medical School approved all procedures, which were also consistent with the ethical guidelines of the National Institutes of Health and the International Association for the Study of Pain. Each effort was made to minimize animal suffering and the number of animals used. The experimenters were blind to viral treatment or drug treatment condition during behavioural testing.

**Animal models.** L$_5$ spinal nerve ligation (SNL) and chronic constriction injury (CCI) models of neuropathic pain in rats and L$_4$ SNL model of neuropathic pain in mice were carried out according to previously published methods[8,9,36,38]. Briefly, for the SNL model, an incision on the lower back was made after the animals were anesthetized with isoflurane. The lumbar transverse process was identified and then removed with the ronguer. The underlying spinal nerve (L$_5$ in rats and L$_4$ in mice) was isolated and ligated with a 3–0 silk thread in rats or 7–0 silk thread in mice. The ligated nerve was then transected distal to the ligature. The skin and muscles were finally closed in layers. For the CCI model, the exposed sciatic nerve was loosely ligated with 3–0 silk thread at four sites with an interval of about 1 mm proximal to trifurcation of the sciatic nerve. The sham groups underwent identical procedures of the SNL or CCI group, but without the ligature of the respective nerve. The complete Freund's adjuvant (CFA)-induced inflammatory pain model in rats was performed as described previously[48,49]. A 100 µl of CFA (1 mg ml$^{-1}$ *Mycobacterium tuberculosis*, Sigma, St. Louis, MO) solution was subcutaneously injected into the plantar side of one hind paw.

**Behavioural analysis.** Mechanical, cold, and thermal tests, as well as locomotor performance were carried out prior to viral injection or surgery and at different time points after surgery. Each behavioural test was carried out at 1 h intervals.

Paw withdrawal thresholds (in rats) or frequencies (in mice) in response to mechanical stimuli (calibrated von Frey filaments) were first measured as described[8,9,36,50]. Briefly, the animal was placed in an individual Plexiglas chamber on an elevated mesh screen. For rats, calibrated von Frey filaments (Stoelting Co., Wood Dale, IL, USA) in log increments of force (0.69, 1.20, 2.04, 3.63, 5.50, 8.51, 15.14 and 26 g) were used to stimulate the plantar surface of the rats' left and right hind paws. The 3.63-g stimulus was used first. If a negative

response occurred, the next larger von Frey hair was applied; if a positive response was seen, the next smaller von Frey hair was applied. The application was terminated when (i) a negative response was seen with the 26-g stimulation or (ii) three stimuli were used after the first positive response. On the basis of a formula provided by Dixon[51], paw withdrawal threshold was calculated by converting the pattern of positive and negative responses to a 50% threshold value. For mice, two calibrated von Frey filaments (0.07 and 0.4 g) were used to stimulate the hind paw for ∼1 s and each stimulation was repeated 10 times to both hind paws with 5 min interval. Paw withdrawal response in each of these 10 applications was represented as a per cent response frequency ((number of paw withdrawals/10 trials) $\times$ 100 = % response frequency), and this percentage was obtained as an indication of the amount of paw withdrawal.

Paw withdrawal latencies to noxious thermal stimulation were then examined with a Model 336 Analgesia Meter (IITC Inc. Life Science Instruments. Woodland Hills, CA), as described previously[8,9,36,50]. In brief, the animal was placed in an individual Plexiglas chamber on a glass plate. A beam of light through a hole in the light box of Model 336 Analgesic Meter through the glass plate was used to stimulate the middle of the plantar surface of each hind paw. The light beam was automatically turned off when the animal withdrew its foot. The paw withdrawal latency was recorded by the length of time between the start of the light beam and the foot withdraw. Each test was repeated five times at 5-min intervals for the paw on each side. To avoid tissue damage to the hind paw, a cut-off time of 20 s was applied.

Paw withdrawal latencies to noxious cold (0 °C) were examined as described previously[8,52,53]. The animal was placed in an individual Plexiglas chamber on the cold aluminium plate, the temperature of which was monitored continuously by a thermometer. The paw withdrawal latency was recorded by the length of time between the placement of the hind paw on the plate and a flinching of the paw. Each test was repeated three times at 10-min intervals for the paw on the ipsilateral side. To avoid tissue damage, a cut-off time of 60 s for rats or 20 s for mice was used.

Locomotor function, including placing, grasping, and righting reflexes, were examined after all behavioural tests based on previously described protocols[8,52,53]. (1) Placing reflex: the placed positions of the hind limbs were slightly lower than those of the forelimbs, and the dorsal surfaces of the hind paws were brought into contact with the edge of a table. Whether the hind paws were placed on the table surface reflexively was recorded; (2) Grasping reflex: After the animal was placed on a wire grid, whether the hind paws grasped the wire on contact was recorded; (3) Righting reflex: when the animal was placed on its back on a flat surface, whether it immediately assumed the normal upright position was recorded. Each trial was repeated five times with 5-min interval and the scores for each reflex were recorded based on counts of each normal reflex.

**Conditioned place preference test.** Conditioned place preference (CPP) test was performed as described previously[31,54]. In brief, the CPP apparatus consisting of two distinct Plexiglas chambers connected with an internal door (MED Associates Inc., St. Albans, VT) was used. The photobeam detectors installed along the chamber walls monitored and automatically recorded the movement of each animal and time spent in each chamber using MED-PC IV CPP software. Three days after SNL or sham surgery, preconditioning was carried out first. The animal was permitted to enter to both chambers for 30 min with the internal door opened. On the third day, the length of time spent in each chamber was recorded after the animal was placed into one chamber with full access to both chambers for 15 min (900 s). Animals spending less than 180 s or more than 720 s in any chamber were ruled out from further testing. The conditioning protocol as described below was carried out on days 4, 5 and 6 post-SNL or sham surgery when the internal door was closed. The animal was injected intrathecally with saline (10 µl) paired with one conditioning chamber in the early morning. The lidocaine (0.8% in 10 µl saline) was injected intrathecally paired with another conditioning chamber in the later afternoon. On day 7 (test day) post-SNL or sham surgery, the animal was placed in one chamber with free access to both chambers. The length of time spent in each chamber was recorded for 15 min. Difference scores were defined as post-conditioning time subtracted from preconditioning time spent in the lidocaine-paired chamber.

**DRG microinjection.** DRG microinjection was performed as described previously[8,9]. Briefly, after the animal was anesthetized with isoflurane, midline incision in the lower lumbar back region was made and the lumbar articular process was exposed and removed. The exposed DRG was injected with viral solution (1–1.5 µl) through a glass micropipette connected to a Hamilton syringe. The pipette was remained 10 min after injection. The animals showing the signs of paresis or other abnormalities were excluded. The injected DRGs were stained with hematoxylin/eosin to examine whether their structure was integrity and whether they contained no visible leukocytes.

**Cell line culture and transfection.** HEK-293 T (ATCC, Manassas, VA) or PC12 (ATCC) cells were cultured in Dulbecco's Modified Eagle's Medium (DMEM; Gibco/ThermoFisher Scientific, Waltham, MA) containing 10% (v/v) fetal bovine serum (FBS; Gibco/ThermoFisher Scientific) at 37 °C in a humidified

incubator with 5% $CO_2$. The plasmids and siRNAs were transfected into the HEK-293T or PC12 cells with Lipofectamine 2,000 (Invitrogen, Carlsbad, CA) according to the manufacturer's instructions and our previous work[9].

**DRG neuronal culture and viral transduction.** Primary DRG neuronal cultures and viral transfection were performed as described[9] with minor modification. In brief, after adult $Dnmt3a^{fl/fl}$ mice euthanized with isoflurane, all DRGs were harvested in cold Neurobasal Medium (Gibco/ThermoFisher Scientific) contained with 10% fetal bovine serum (JR Scientific, Woodland, CA), 100 units ml$^{-1}$ Penicillin, and 100 μg ml$^{-1}$ Streptomycin (Quality Biological, Gaithersburg, MD). The DRGs were then treated with enzyme solution (5 mg ml$^{-1}$ dispase, 1 mg ml$^{-1}$ collagenase type I in Hanks' balanced salt solution (HBSS) without $Ca^{2+}$ and $Mg^{2+}$ (Gibco/ThermoFisher Scientific)). After trituration and centrifugation, the dissociated neurons resuspended in mixed Neurobasal Medium were plated in a six-well plate coated with 50 μg ml$^{-1}$ poly-D-lysine (Sigma). The cells were incubated at 95% $O_2$, 5% $CO_2$ and 37 °C. On the second day, 0.5–1 μl of virus (titre ≥ $1 \times 10^{12}$) was added to each well. Three days later, the neurons were harvested for RNA extraction as described below.

**Reverse transcription (RT)-PCR.** Quantitative real-time RT-PCR was carried out as described[9,55]. To achieve enough RNA, two unilateral rat DRGs or four unilateral mouse DRGs were pooled together. Total RNA from the tissues or the cultured samples was extracted by the Trizol method (Invitrogen/ThermoFisher Scientific, Grand Island, NY), treated with overdose of DNase I (New England Biolabs, Ipswich, MA), and reverse-transcribed using the ThermoScript reverse transcriptase (Invitrogen/ThermoFisher Scientific), oligo (dT) primers or specific RT-primers (Supplementary Table 3). Template (1 μl) was amplified by real-time PCR using the primers listed in Supplementary Table 3 (Integrated DNA Technologies). Each sample was run in triplicate in a 20 μl reaction with 250 nM forward and reverse primers, 10 μl of SsoAdvanced Universal SYBR Green Supermix (Bio-Rad Laboratories, Hercules, CA) and 20 ng of cDNA. Reactions were carried out on a BIO-RAD CFX96 real-time PCR system. $Gapdh$ or $Tuba-1a$ was used as an internal control for normalization, as they have been demonstrated to be stable even after peripheral nerve injury insult[9,56]. Ratios of ipsilateral-side mRNA levels to contralateral-side mRNA levels were calculated using the ΔCt method ($2^{-\Delta\Delta Ct}$).

Single-cell real-time RT-PCR was performed as described previously[9,55]. Briefly, the freshly cultured DRG neurons were prepared as described before. Four hours after plating, under an inverted microscope fit with a micromanipulator and microinjector, a single living large (> 35 μm), medium (25–35 μm), and small (<25 μm) DRG neuron was harvested in a PCR tube with 5-10 μl of cell lysis buffer (Signosis, Sunnyvale, CA). After centrifugation, the supernatants were harvested and divided into two PCR tubes for $Oct1$ and $Dnmt3a$ genes. The remaining real-time RT-PCR procedure was performed based on the manufacturer's instructions with the single-cell real-time RT-PCR assay kit (Signosis). All primers used are listed in Supplementary Table 3.

**Plasmid constructs and virus production.** By using the SuperScript III One-Step RT-qPCR System with the Platinum Taq High Fidelity (Invitrogen/Thermo-Fisher Scientific) and the primers (Supplementary Table 3), rat full-length $Dnmt3a$ cDNA or $Oct1$ cDNA was synthesized and amplified from total RNA of rat DRG. A $Dnmt3a$ shRNA duplex corresponding to bases 2576-2594 from the open reading frame of rat $Dnmt3a$ mRNA (GenBank accession number NM_001003958) was designed. A mismatch shRNA with a scrambled sequence and no known homology to a rat gene (scrambled shRNA) was used as a control. shRNAs were synthesized and were amplified by using the primers listed in Supplementary Table 3. Fragments harbouring $Dnmt3a$, $Oct1$, $Dnmt3a$ shRNA, and scrambled shRNA were ligated into pro-viral plasmids using using AgeI and XbaI restriction sites. The resulting vectors expressed the genes under the control of the cytomegalovirus promotor. AAV5 viral particles carrying the cDNA were produced at the UNC Vector Core (Chapel Hill, NC). AAV5-GFP and AAV5-Cre were purchased from UNC Vector Core. HSV-GFP and HSV-Dnmt3a were provided by Dr. Eric J Nestler. $Oct1$ siRNA (Catalogue number: sc-36120) and its negative control siRNA (catalogue number: sc-37007) were purchased from Santa Cruz Biotechnology, Inc. (Dallas, TX).

**Immunohistochemistry.** After animals were deeply anesthetized with isoflurane, they were perfused with 100–300 ml of 4% paraformaldehyde in 0.1 M phosphate buffer (pH 7.4). The L4 and L5 DRG were harvested, postfixed at 4 °C for 4 h, and cryoprotected in 30% sucrose overnight. The tissues were sectioned at the thickness of 20 μm on a cryostat. After being blocked with PBS containing 5% goat serum and 0.3% Triton X-100 for 1 h at 37 ºC, the sections were incubated overnight at 4 °C with a mixture of rabbit anti-DNMT3a (1:500, Santa Cruz Biotechnology, Inc.) and mouse anti-NeuN (1:50, Genetex, Irvine, CA), mouse anti-glutamine synthetase (GS, 1:200, EMD Millipore, Darmstadt, Germany), mouse anti-calcitonin gene-related peptide (CGRP, 1:50, Abcam, Cambridge, MA), biotinylated isolectin B4 (IB4, 1:100, Sigma), mouse anti-neurofilament-200 (NF200, 1:800, Sigma) or mouse anti-Kv1.2 (Kcna2, 1:200, NeuroMab, Davis, CA). The sections were then incubated with a mixture of goat anti-rabbit IgG conjugated

with Cy3 (1:200) and donkey anti-mouse IgG conjugated with Cy2 (1:200, Jackson ImmunoResearch) or avidin-FITC (1:200, Sigma) for 1 h at room temperature. Control experiments included omission of the primary antiserum and substitution of normal mouse or rabbit serum for the primary antiserum. The sections were finally mounted using VectaMount permanent mounting medium (Vector Laboratories) and coverslipped with Vectashield plus 4′, 6-diamidino-2-phenylindole (DAPI) mounting medium (Vector Laboratories). All images were observed using a Leica DMI4000 fluorescence microscope and captured with a DFC365FX camera (Leica, Germany). Single-, double- or triple-labelled cells were calculated manually or using NIH Image J Software.

**Western blotting.** Bilateral L4 and L5 DRGs and L5 spinal cord were collected. To achieve a high protein concentration, two unilateral rat DRG or four unilateral mouse DRG were pooled together. The tissues were homogenized with ice-cold lysis buffer, which contained 0.25 M sucrose, 10 mM Tris, 2 mM $MgCl_2$, 5 mM EGTA, 1 mM phenylmethylsulfonyl fluoride, 1 mM DTT, and 40 μM leupeptin. After the crude homogenate was centrifuged at 4 °C for 15 min at 1,000g, the supernatants were collected for cytosolic proteins and the pellets for nuclear proteins. After measuring protein concentration, the samples were heated for 5 min at 99 °C and loaded onto a 4% stacking/7.5% separating SDS-polyacrylamide gel (Bio-Rad Laboratories). The proteins were then electrophoretically transferred onto a nitrocellulose membrane (Bio-Rad Laboratories). After being blocked with 3% nonfat milk in Tris-buffered saline containing 0.1% Tween-20 for 1 h, the membranes were then incubated with following primary antibodies overnight. These antibodies included rabbit anti-DNMT3a (1:500, Cell Signaling Technology, Danvers, MA), rabbit anti-DNMT3b (1:500, Santa Cruz Biotechnology, Inc.), rabbit anti-DNMT1 (1:1,000, Cell Signaling Technology), rabbit anti-TET1 (1:500, Abcam), rabbit anti-TET2 (1:500, EMD Millipore), rabbit anti-TET3 (1:500, Thermal Scientific, Grand Island, NY), rabbit anti-OCT1 (1:500, Abcam), rabbit anti-H3 (1:2,000, Cell Signaling Technology), rabbit anti-Kv1.1 (Kcna1, 1:200, NeuroMab), rabbit anti-Kv1.2 (Kcna2, 1:200, NeuroMab), rabbit anti-Kv1.4 (Kcna4, 1:200, NeuroMab), rabbit anti-p-ERK1/2 (1:200, Cell Signaling Technology), rabbit anti-ERK1/2 (1:1,000, Cell Signaling Technology), rabbit anti-GFAP (1:1,000, Cell Signaling Technology), rabbit anti-GAPDH (1:2,000, Santa Cruz Biotechnology, Inc.), rabbit anti- β-actin (1:1,000, Cell Signaling Technology) and rabbit anti-α-tubulin (1:2,000, Santa Cruz Biotechnology, Inc.). The proteins were detected by horseradish peroxidase–conjugated anti-mouse or anti-rabbit secondary antibody (1:3,000, Jackson ImmunoResearch) and visualized by western peroxide reagent and luminol/enhancer reagent (Clarity Western ECL Substrate, Bio-Rad) and exposure using the ChemiDoc XRS System with Image Lab software (Bio-Rad). The intensity of blots was quantified with densitometry using Image Lab software (Bio-Rad). The average blot density from the control/naive groups was set as 100%. The relative density values from time points or the treated groups were determined by dividing the optical density values from these groups by the average value of the control/naive groups after each was normalized to the corresponding histone H3 (for nucleus proteins), GAPDH, α-tubulin, or β-actin (for cytosolic proteins).

**Chromatin immunoprecipitation (ChIP) assay.** The ChIP assays were carried out using the EZ ChIP Kit (Upstate/EMD Millipore, Darmstadt, Germany) as described previously[9]. The crude homogenate from the DRG was crosslinked with 1% formaldehyde at room temperature for 10 min. The reaction was stopped by adding glycine (0.25 M). After centrifugation, the pellet was collected and lysed in SDS lysis buffer containing protease inhibitor cocktail. The lysis was sonicated until the DNA was broken into fragments with a mean length of 200–1,000 bp. The samples were first pre-cleaned with protein G agarose and then subjected to immunoprecipitation overnight with 2 μg of rabbit antibodies against OCT1 (Abcam) or DNMT3a (Abcam) or with 2 μg of normal rabbit serum overnight at 4 °C. The 10–20% of the sample for immunoprecipitation was used as an input (a positive control). After purification, the DNA fragments were amplified using PCR/Real-time PCR with the primers listed in Supplementary Table 3.

**Luciferase assay.** The 885-bp fragment from the $Dnmt3a$ gene promotor region (including OCT1-binding motif) and the 696-bp fragment from $-622$ to $+74$ bp of the $Kcna2$ gene promotor and 5′-untranslated region were amplified by PCR from genomic DNA with the primers (Supplementary Table 3) to, respectively, construct the $Dnmt3a$ and $Kcna2$ gene reporter plasmids. Vectors containing mutations at the $-482$, $-457$, $-444$ or $-440$ CpG site were achieved by deleting the corresponding C using the MutanBEST kit (Takara, Berkeley, CA) to achieve site-directed mutagenesis. The PCR products were ligated into the pGL3-Basic vector (containing $firefly$ luciferase reporter gene, Promega, Madison, WI) using the SmaI and HindIII restriction sites. DNA sequencing was carried out to verify the sequences of recombinant clones. HEK-293T cells were cultured as described above. One day after the cells were plated on 12-well plate, the cells were transfected with 40 ng of the pRL-TK plasmid (as a control containing $renilla$ luciferase reporter gene, Promega) alone or plus 1 μg of the pGL3-Basic vectors using Lipofectamine 2000 (Invitrogen), according to the manufacturer's instructions. The wells were divided into different groups as indicated. Two days after transfection, the cells were collected in passive lysis buffer. Approximately

40 μl of supernatant was used to measure the luciferase activity with the Dual-Luciferase Reporter Assay System (Promega). Independent transfection experiments were repeated for three times. The relative reporter activity was calculated after normalization of the *firefly* activity to *renilla* activity.

**Bisulfite sequencing.** Seven days after SNL or sham surgery, L5 DRG on the ipsilateral side was collected for DNA extraction. The treatment of genomic DNA with bisulfite was carried out using the EZ DNA Methylation-Gold kit (ZYMO Research, Irvine, CA), according to the manufacturer's instructions. The region of the *Kcna2* gene promotor and 5′UTR from the −540 to +500 bp site consisted of 65 CpG sites and were amplified. For the pyrosequence study, 4 pairs of primers of *Kcna2* modified with 5′-Biotin (Supplementary Table 3) were used to amplify the bisulfite DNA. The master mix of the binding buffer, streptavidin-sepharose beads, and PCR products were then prepared for the binding reaction in a 96-well plate. The pyrosequence was performed using a PSQ HS96 (Biotage, Charlotte, NC) to determine percentage methylation at each CpG site. For the clone-sequencing study, the same primers without biotin modification were used in PCR amplification. The PCR products were purified and subcloned into pMD T-19 (Takara). After an overnight bacterial culture, 20 subclones from each PCR assay were subjected to direct sequencing.

**Whole-cell patch clamp recording.** Five weeks after viral microinjection into the DRG, the freshly dissociated DRG culture was prepared. Whole-cell patch clamp recording of DRG neurons was carried out 4–10 h after the plating. The micropipettes' electrode resistances ranged from 3 to 5 MΩ. The neurons were voltage-clamped with an Axopatch-700B amplifier (Molecular Devices, Sunnyvale, CA). The intracellular pipette solution consisted of (in mM) potassium gluconate 120, KCl 20, MgCl$_2$ 2, EGTA 10, HEPES 10 and Mg-ATP 4 (pH 7.3 with KOH, 310 mOsm). To diminish the Na$^+$ and Ca$^{2+}$ component in voltage-gated potassium current, an extracellular solution contained (in mM) choline chloride 150, KCl 5, CdCl$_2$ 1, CaCl$_2$ 2, MgCl$_2$ 1, HEPES 10 and glucose 10 (pH 7.4 with Tris base, 320 mOsm). Signals were filtered at 1 kHz and digitized using a DigiData 1550 with pClamp 10.4 software (Molecular Devices). Series resistance was compensated by 60–80%. Through reading the value for whole-cell capacitance compensation directly from the amplifier, cell membrane capacitances were achieved. To eliminate leak current contribution, we carried out an online P/4 leak subtraction. We stored all data into a computer by a DigiData 1,550 interface and analysed them by the pCLAMP 10.4 software package (Molecular Devices).

The neurons were also current-clamped to record the action potential (AP). The extracellular solution contained (in mM) NaCl 140, KCl 4, CaCl$_2$ 2, MgCl$_2$ 2, HEPES 10 and glucose 5, with pH adjusted to 7.38 by NaOH. The intracellular pipette solution was composed of (in mM) KCl 135, Mg-ATP 3, Na$_2$ATP 0.5, CaCl$_2$ 1.1, EGTA 2 and glucose 5; pH was adjusted to 7.38 with KOH and osmolarity adjusted to 300 mOsm with sucrose. Three to five min after a stable recording was first achieved, the resting membrane potential was recorded. We injected current pulses (100–1,400 pA, 200 ms) into the soma through the recording electrode to evoke the AP. During the current injection, the membrane potential was kept at the existing resting membrane potential. We defined the injection current threshold as the minimum current required to evoke the first AP and the AP threshold as the first point on the rapid rising phase of the spike at which the change in voltage exceeded 50 mV ms$^{-1}$. The AP amplitude was obtained between the peak and the baseline. The membrane input resistance for each neuron was achieved from the slope of a steady-state I–V plot in response to a series of hyperpolarizing currents, 200-ms duration delivered in steps of 100 pA from 200 pA to −2,000 pA. We measured the after-hyperpolarization amplitude between the maximum hyperpolarization and the final plateau voltage and the AP overshoot between the AP peak and 0 mV. These recorded data were stored on a computer by a DigiData 1,550 interface and were analysed by the pCLAMP 10.4 software package (Molecular Devices). The experiments were carried out at room temperature.

**Statistical analysis.** For *in vitro* experiments, the cells were evenly suspended and then randomly distributed in each well tested. For *in vivo* experiments, the animals were distributed into various treatment groups randomly. All of the results were given as means ± s.e.m. The data were statistically analysed with two-tailed, unpaired Student's *t*-test and a one-way or two-way ANOVA. When ANOVA showed significant differences, pairwise comparisons between means were tested by the *post hoc* Tukey method (SigmaStat, San Jose, CA). No statistical methods were used to predetermine sample sizes, but our sample sizes are similar to those reported previously in the field[8,9,36,48,55]. Significance was set at P < 0.05.

**Data availability.** The data sets generated during and/or analysed during the current study are available from the corresponding author on reasonable request.

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

## Acknowledgements

We thank Dr Richard Jude Samulski (University of North Carolina) for providing the AAV5 plasmid, Dr Hongjun Song (Johns Hopkins University School of Medicine) for providing the shRNA plasmid and Han-Rong Weng (University of Georgia College of Pharmacy) for electrophysiological data analysis. This work was supported by NIH grants (R01NS094664, R01NS094224, R01DA033390, NS 072206 and U01HL117684) to Y.X.T., by the National Natural Science Foundation of China (81671094) to Z.L., by 863 program of China (2015AA020913) to J.Y.Z. and by a NIH research Fellowship (F31NS092310) to B.M.L.

## Author contributions

Y.-X.T. conceived the project and supervised all experiments. J.-Y.Z., L.L., X.G., Z.L., S.W. and Y.-X.T. assisted with experimental design. J.-Y.Z., L.L., Z.L., S.W., L.S., F.E.A., K.M., S.J. and B.M.L. carried out animal surgery and molecular, biochemical and behavioural experiments. X.G. performed patch clamp recording. J.F. and E.J.N. generated HSV and *Dnmt3a*^{fl/fl}. J.-Y.Z., L.L., X.G., Z.L., S.W., A.B. and Y.-X.T. analysed the data. Y.-X.T. wrote the manuscript. All authors read and discussed the manuscript.

## Additional information

**Competing financial interests:** The authors declare no competing financial interests.

