## [Peer Review File · Nature Communications]

Reviewers' comments:

Reviewer #1 (Remarks to the Author):

The manuscript by Zhao et al. "Contribution of DNMT3a to neuropathic pain genesis by downregulating Kcna2 in primary afferent neurons" describes the contribution of DNMT3a to the development of neuropathic pain. The authors describe the upregulation of DNMT3a in DRGs following SNL, provide the evidence that this upregulation is due to the increase of the transcription factor OCT1 and show that inhibition of DNMT3a by shRNA could attenuate neuropathic pain states while increase in DNMT3a using AAV5 could induce neuropathic pain symptoms. Finally, the authors intend to show that DNMT3a regulates neuropathic pain states via downregulation of Kv1.2, but this part of the study is less convincing.

Overall this is a large and important piece of work. It is rather novel as so far no one has directly shown that DNMT3a could modulate chronic pain states. The methodology is suitable and the statistical analysis seems appropriate. However, while the contribution of DNMT3a to the development of neuropathic pain is rather compelling, the proposed mechanism is not entirely convincing. The authors suggest that DNMT3a contributes to the development of neuropathic pain by silencing Kcna2 in DRGs, by increasing DNA methylation levels in the Kcna2 sequence. However, according to the authors, only 4 CpG sites located within DNMT3a binding regions in the Kcna2 sequence show changes in DNA methylation following SNL and out of these, only 1 CpG site shows reversal of the SNL induced changes by DNMT3a KO. Does this mean that DNMT3a is only responsible for SNL induced changes in DNA methylation at 1 CpG site on the Kcna2 sequence? And how would this be relevant to the expression of the Kv1.2 channel? This is I think the weakest point of the study. I would suggest changing the discussion in places to indicate that the link demonstrated is tenuous. For example p.13: "This increase [in DNMT3a] led to an elevation in the level of DNA methylation at the CpG sites...". It would be wiser to talk about correlation since DNMT3a only modulated DNA methylation at 1 site. Finally, the authors suggest that DNMT3a could be a novel target for neuropathic pain management. However, the idea of targeting an enzyme universally expressed and important for the regulation of global gene expression in order to manage chronic pain states is rather controversial and requires a proper discussion.

Other concerns

1. In Fig.1: the authors claim that DNMT3a is only expressed in neurones in DRGs. They quote the lack of double labelling with GS as a sign of no expression in satellite glia cells. But unless the authors also use a nuclear marker to locate the nuclei of satellite cells, it is impossible to conclude that DNMT3a is not expressed by this cell type.
2. I have some concerns regarding the Western Blot data throughout the manuscript. First of all, the loading controls, measured by the H3 signal, are often very poor. For example in Fig.2a, 2b, 5h, Supplemental Fig.1f, Fig.3a, c, f; I have no idea how the authors were able to measure accurately the intensity of the signal attributed to a single well. This is really worrying. My second point concerns the quantification of the signal: why do the control values never display an error bar, for standard error of the mean, while the other groups do? I do understand that other groups are expressed as a ratio of the control group, but this

does not prevent the control groups from having error bars! And this makes me wonder how the differences were statistically evaluated... The qPCR data presented in this manuscript always show the error bars for the control groups so I do not understand why this is not the case for the Western data.

3. References to DNA methylation as a major epigenetic mechanism are insufficient. The authors have quoted a review of their own or old references.

4. Fig.3: where is Fig.3a quantified? The authors talk about significant increase in binding activity but only mention 2.75-fold increase. Quantification and statistical details must be given.

5. Fig.4 is not very well presented and should be clarified. At least the legend should not be spread across 3 graphs for each row. Are 2 groups missing in graph 4l?

6. Fig.3b-d are very important as they are showing the effects of the shRNA on Dnmt3a expression. These should be in the main manuscript. If I am correct, the shRNA by itself had no effect at all on DNMT3a protein levels but reduced the mRNA by 50%, which is a rather large effect (Supplemental Fig.3d). This needs to be discussed.

7. Please show the results of shRNA and virus injection on locomotor function in Supplementary data (instead of mentioning as "data not shown").

8. Contrary to the findings reported in the manuscript, others have previously reported that DNMT3a was upregulated in the superficial dorsal horn following CFA induced ankle joint inflammation (Tochiki et al. 2012). This should be discussed.

9. Please provide F values and p values for all ANOVA throughout the manuscript.

10. Does OCT1 bind other targets which could be relevant to nociceptive processing in DRGs? Please discuss.

11. Often in the manuscript the data is presented normalized to contra. I think that the authors are missing the opportunity to display the contra data which could be very interesting. Why not normalizing to sham?

12. The style is sometimes unclear and there are quite few typos. E.g.: in the "Figure Legends", the Tukey test is referred to throughout as the "Turkey test".

13. Fig.2a and 2b should be on the same scale (Y-axis).

Reviewer #2 (Remarks to the Author):

The manuscript by Zhao et al. reports novel results demonstrating that nerve injury-induced upregulation of the methyltransferase DNMT3a is specifically involved in the downregulation of the DRG potassium channel *Kcna2*, which induces DRG hyperexcitability and neuropathic pain. The authors make a very strong case for this conclusion by measuring multiple parameters (RNA, protein, function, behavior) and assessing several complementary conditions (sham, SNL, shRNA, fl/fl + Cre, overexpression, etc.). Furthermore, they characterized the injury-induced upregulation of OCT1, the transcription factor that regulates the DNMT3a promoter. In the end, the data strongly suggest that OCT1 promotes DNMT3a expression, which in turn selectively methylates the *Kcna2* promoter and this, through an unknown mechanism, silences expression of the potassium channel. To the best of my knowledge, this is the first study that demonstrates in detail how a specific epigenetic modification plays a key role in the development of neuropathic pain through its effect on

the expression of a particular potassium channel previously implicated in this malady.

Overall, the manuscript is well written, the approach is extensive and thorough, the quality of the experiments is excellent and the conclusions follow logically from the experimental results. Also, the methods provide sufficient experimental detail for possible replication. There are, however, several important issues that the authors need to address to clarify the results, their interpretation and the conclusions.

Major Comments

1) The authors conclude that DNMT3a upregulation selectively downregulates Kcna2 (compared to Kcna1 and Kcna4) and that this dysregulation is mainly responsible for the increased excitability of large- and medium-size DRG neurons. However, it is clear that SNL also downregulates the Kcna1 and Kcna4 transcripts in the DRG (Fig. 6a/c) and that the largest change induced by the DNMT3a treatment was to increase the excitability of small-size DRG neurons (Fig. 8i). This is intriguing because Kcna2 is poorly expressed in small-size DRG neurons. An expanded critical discussion of this interesting and important observation would add impact to this study (including possible explanations).

2) Previously, the authors reported that, following nerve injury, upregulation of the Kcna2 long non-coding antisense RNA attenuates expression of Kcna2. However, the exact mechanism underlying this outcome is not fully understood. Toward solving this mechanism, the authors refer to a previous study, which showed that the long non-coding antisense RNA Tsix specifically binds to DNMT3a to mediate de novo DNA methylation at the Xist promoter and induce repression of Xist. Although this is tantalizing, the authors only speculate that a similar mechanism might underlie the silencing effect of the Kcna2 long non-coding antisense RNA. Unless the authors have experimental evidence or there is established generality for the proposed mechanism, I suggest to tone down/delete this conclusion.

Instead, critically discussing point #1 above might be more beneficial to readers wondering about intriguing experimental observations reported in this manuscript.

Minor Comments

1. Following SNL, does the OCT1 mRNA increase in the DRG?
2. Line 35: It would be more accurate to say "attenuating Kcna2 expression" than "silencing DRG Kcna2".
3. Line 26: insert "the" between "of" and "dorsal root ganglion".
4. Please consider revising the sentence running from line 38 to line 40.
5. Throughout the figure legends "Tukey" is misspelled ("Turkey"). Please revise.

Reviewer #3 (Remarks to the Author):

The manuscript by Zhao et al. explores the role of DNMT3a in regulating voltage-gated potassium channels in the DRG of neuropathic rodents and the subsequent experience of pain. The main subject of these investigations, Kv1.2, and related K-channels are very well established to contribute to allodynia in nerve injury models. What is novel here is the establishment of a pathway leading from nerve injury to down-regulation of one target of DNMT3a. There are other genes known to be regulated by methylation in peripheral, DRG and spinal tissue in various pain models. This is a particularly strong story for one gene. Overall, the experimental approach is solid and multidisciplinary, and the methodology rigorous.

General comments:

1. Reflexive measures of nociception can no longer be considered to be adequate tests of the pain-like state that may be present in this and other pain models. Especially for a higher impact journal, tests of spontaneous pain such as CPP, vocalization, etc. or functional effects of the injury, e.g. gait, should be included. Sensitization alone is inadequate for the purpose of translation.
2. The authors measure the effects of DNMT3a on one gene, but there is no reason to think that many genes are not under the same enzyme's control, including others that might contribute to pain. One of the main characteristics of epigenetic processes is that they generally are not single-gene selective.
3. The authors make use of decitabine/zebularine/5azadC. Mechanistically, this drug requires cell proliferation in the tissues/cells of interest and should only target DNMT1. Why wasn't another drug used? (For example, RG101 is more specific and specifically targets DNMTa/b). The inclusion of the knockdown data helped narrow down the function to DNMT3A but, in general, decitabine should not be working on DNMT3a.
4. The authors use mouse and rat SNL models throughout the manuscript. Could the authors comment on whether they observed any species difference in instances where both species were examined for the same parameters?
5. The discussion is simply a repetition of the "results" section and offers no real insight into the significance of the presented data or their fit with the current state of knowledge. Furthermore, both the introduction and discussion sections are missing key pieces of published literature from the field of pain epigenetics.
6. Line 63/64 and 278/279: The claim that "DNMT3A is a new pain target" is overstated. DNMTs are widely and ubiquitously expressed, and while it may be tempting to over-interpret the findings and claim "therapeutic potential", the findings presented in the manuscript offer very limited data for such broad statements. Chemotherapeutics targeting methyltransferases have very severe side effects.

Figures:

There are currently 8 (very busy) figures that are frustrating to go through. Perhaps the authors could consider revising the content and organization of their panels and/or moving some of these data to supplemental information. For instance, figure 1 does not add much

to the main manuscript and would be an ideal candidate for supplementary data.

Figure 2: The DNMT3b bands, particularly in the sham group, are very "fuzzy" and I have doubts whether they could be quantified with any level of reliability.

Figure 4: This figure is extremely confusing due to the placement of the symbol legends. For each of the 4 rows of data, please have a dedicated space for all the symbol legends so that the reader is spared going back and forth through the panels to decipher the different symbols.

The point to point responses to the reviewers' comments:

Response to the Reviewer #1 comments:

The manuscript by Zhao et al. "Contribution of DNMT3a to neuropathic pain genesis by downregulating Kcna2 in primary afferent neurons" describes the contribution of DNMT3a to the development of neuropathic pain. The authors describe the upregulation of DNMT3a in DRGs following SNL, provide the evidence that this upregulation is due to the increase of the transcription factor OCT1 and show that inhibition of DNMT3a by shRNA could attenuate neuropathic pain states while increase in DNMT3a using AAV5 could induce neuropathic pain symptoms. Finally, the authors intend to show that DNMT3a regulates neuropathic pain states via downregulation of Kv1.2, but this part of the study is less convincing.

Overall this is a large and important piece of work. It is rather novel as so far no one has directly shown that DNMT3a could modulate chronic pain states. The methodology is suitable and the statistical analysis seems appropriate. However, while the contribution of DNMT3a to the development of neuropathic pain is rather compelling, the proposed mechanism is not entirely convincing. The authors suggest that DNMT3a contributes to the development of neuropathic pain by silencing Kcna2 in DRGs, by increasing DNA methylation levels in the Kcna2 sequence. However, according to the authors, only 4 CpG sites located within DNMT3a binding regions in the Kcna2 sequence show changes in DNA methylation following SNL and out of these, only 1 CpG site shows reversal of the SNL induced changes by DNMT3a KO. Does this mean that DNMT3a is only responsible for SNL induced changes in DNA methylation at 1 CpG site on the Kcna2 sequence? And how would this be relevant to the expression of the Kv1.2 channel? This is I think the weakest point of the study. I would suggest changing the discussion in places to indicate that the link demonstrated is tenuous. For example p.13: "This increase [in DNMT3a] led to an elevation in the level of DNA methylation at the CpG sites...". It would be wiser to talk about correlation since DNMT3a only modulated DNA methylation at 1 site. Finally, the authors suggest that DNMT3a could be a novel target for neuropathic pain management. However, the idea of targeting an enzyme universally expressed and important for the regulation of global gene expression in order to manage chronic pain states is rather controversial and requires a proper discussion.

We highly appreciate the reviewer's suggestion. We carried out 3 experiments: 1. Performing bisulfite pyro-sequencing to confirm the bisulfite clone-sequencing data, 2. Increasing the number of subclones (total 20 subclones/group) in clone sequencing, and 3. Determining whether methylation sites (-457 and -444) are relevant to the promoter activity of *Kcna2* gene.

To further confirm the results from the bisulfite clone-sequencing, we used the bisulfite pyro-sequencing assay and verified the increases in DNA methylation at -457 and -444 CpG sites, although an increase was detected at an additional -482 CpG site, in the injured DRG on day 7 post-SNL (Fig. 7d). We also demonstrated that SNL-induced increases in DNA methylation at -457 and -444 CpG sites are DNMT3a-dependent (Fig. 7e,f). Furthermore, we revealed that DNMT3a overexpression markedly reduced *Kcna2* promoter activity, which could be blocked by the deletion of both -457 and -444 CpG sites (Supplemental Fig. 4a,b). It is very likely that the increased DNA methylation at

these two sites within the promoter of the *Kcna2* gene may block the binding of the transcription factors to these regions, resulting in silencing of the *Kcna2* gene.

It should be pointed out that DNA methylation was examined only from the -540 to +500 bp of the *Kcna2* gene (covering the promoter and 5-UTR region of the *Kcna2* gene) consisted of 65 CpG sites in the present study. It is possible that SNL-induced increases in DNMT3a-triggered DNA methylation occurs at other CpG sites of the *Kcna2* gene. These sites may also be involved in the SNL-induced decrease in *Kcna2* expression in the injured DRG.

These points have been added in the Results Part (page 11, lines 12-23). We have re-written the discussion part (page 13, lines 13-14; page 15, lines 10-23; page 16, line 1).

Finally, we thank the reviewer for his/her concern on DNMT3a as a novel target for neuropathic pain management. The proper discussion has been given on page 17, lines 19-23; page 18, lines 1-2.

Other concerns

1. In Fig.1: the authors claim that DNMT3a is only expressed in neurons in DRGs. They quote the lack of double labelling with GS as a sign of no expression in satellite glia cells. But unless the authors also use a nuclear marker to locate the nuclei of satellite cells, it is impossible to conclude that DNMT3a is not expressed by this cell type.

We have carried out the triple staining of DNMT3a, GS, and DAPI. DNMT3a is not expressed in cellular nuclei of GS-labeled cells (Fig. 1b). We revised the manuscript on page 4, line 13; page 24, lines 12-15; page 31, lines 3-4.

2. I have some concerns regarding the Western Blot data throughout the manuscript. First of all, the loading controls, measured by the H3 signal, are often very poor. For example in Fig.2a, 2b, 5h, Supplemental Fig.1f, Fig.3a, c, f; I have no idea how the authors were able to measure accurately the intensity of the signal attributed to a single well. This is really worrying. My second point concerns the quantification of the signal: why do the control values never display an error bar, for standard error of the mean, while the other groups do? I do understand that other groups are expressed as a ratio of the control group, but this does not prevent the control groups from having error bars! And this makes me wonder how the differences were statistically evaluated... The qPCR data presented in this manuscript always show the error bars for the control groups so I do not understand why this is not the case for the Western data.

According to the reviewer's suggestions, the quality of the Western blot data, especially the loading controls, has been improved.

In order to achieve great separation of the proteins, histone H3 protein always stays near bottom of the SDS-polyacrylamide gel as it is a small protein (15 KDa). This often causes adjacent bands from adjacent wells to connect toward each other. In addition, we sometimes have to load a rather large amount of total protein to obtain the signals for

some targeted proteins that are inherently poorly expressed in the tissues (e.g., DNMT3b). Given histone H3 is expressed richly in the tissues, such loading led to the high intensity bands for H3 even if the exposure time and ECL staining time are reduced at the minimum level. This factor also caused adjacent bands to connect toward each other. However, the margins of each band at the connecting sites can still be recognized, particularly under higher magnification. Thus, the band intensity could be measured accurately.

We are sorry that we did not make clear how the Western blot data were quantified. The intensity of blots was quantified with densitometry using Image lab software. The blot density from the control/naïve group was set as 100%. The relative density values from other time points or the treated groups were determined by dividing the optical density values from these time points or the treated groups by the value of the control/naïve group after each was normalized to the corresponding histone H3 (for nucleus proteins), GAPDH, α -tubulin, or β -actin (for cytosolic proteins). The experiments were repeated at least 3 times/time point or treatment, but the blot density from the control/naïve group per experiment was always set as 100%. Thus, the control/naïve group does not have error bars. This qualified strategy has been carried out in previous studies (Zhao X et al., Nature Neurosci 2013; Xu J et al., J Clin Inves 2014). These points have been added in the revision (page 25, lines 19-23; page 26, lines 1-2).

In contrast, the quantification for RT-PCR analysis is different from the Western blot analysis. Ratios of ipsilateral-side mRNA levels to contralateral-side mRNA levels were calculated using the Δ Ct method ($2^{-\Delta\Delta C_t}$) after each PCR cycle was normalized to a housekeeping gene control (e.g., *Gapdh* or *Tuba-1a*). The experiments were repeated at least 3 times/time point or treatment. Thus, the control/naïve group has an error bar.

3. References to DNA methylation as a major epigenetic mechanism are insufficient. The authors have quoted a review of their own or old references.

We have added two review articles (Turek-Plewa J and Jagodzinski PP, Cell Mol Biol Lett 2005; Poetsch AR and Plass C, Cancer treat Rev 2011), in which DNA methylation as a major epigenetic mechanism was discussed. (page 3, line 12).

4. Fig.3: where is Fig.3a quantified? The authors talk about significant increase in binding activity but only mention 2.75-fold increase. Quantification and statistical details must be given.

Quantification and statistical details have been added in Fig. 3a.

5. Fig.4 is not very well presented and should be clarified. At least the legend should not be spread across 3 graphs for each row. Are 2 groups missing in graph 4l?

The legend in Fig. 4 was rewritten. Two missed groups in the previous Fig. 4L have been added in the revised Fig. 4 (Fig. 4k).

6. *Supplemental Fig.3b-d are very important as they are showing the effects of the shRNA on Dnmt3a expression. These should be in the main manuscript. If I am correct, the shRNA by itself had no effect at all on DNMT3a protein levels but reduced the mRNA by 50%, which is a rather large effect (Supplemental Fig.3d). This needs to be discussed.*

We agree with the reviewer. Fig. 4 has been re-organized in the revision.

Supplemental Fig. 3b-d and 3e-g in the previous version have been moved to Fig. 4a,b and 4g,h, respectively, in the main manuscript.

We observed that both injections of AAV5-DNMT3a shRNA into rat DRG and of AAV5-Cre into the *Dnmt3a*^{fl/fl} mouse DRG decreased basal level of DNMT3a mRNA, but did not alter its protein expression in the sham group. The reason for no effect of viral injection into DRG on basal DNMT3a protein expression is unknown, but the fact is that the remaining DNMT3a mRNA after its knockdown may have highly translational efficacy and maintains normal level of basal DNMT3a protein in the injected DRG. This conclusion is further supported by our behavioral observations that viral injected sham rats or mice displayed normal responses to mechanical, thermal, and cold stimuli. These points have been discussed in the Discussion part (page 14, lines 11-18).

7. *Please show the results of shRNA and virus injection on locomotor function in Supplementary data (instead of mentioning as "data not shown").*

Locomotor function after shRNA and virus injection is shown in supplemental Table 1.

8. *Contrary to the findings reported in the manuscript, others have previously reported that DNMT3a was upregulated in the superficial dorsal horn following CFA induced ankle joint inflammation (Tochiki et al. 2012). This should be discussed.*

We have discussed this point (page 14, lines 7- 10).

9. *Please provide F values and p values for all ANOVA throughout the manuscript.*

We have added all F values and p values in figure legends.

10. *Does OCT1 bind other targets which could be relevant to nociceptive processing in DRGs? Please discuss.*

This study focused on how nerve injury up-regulated DNMT3a gene expression. We have discussed that, in addition to OCT1, other transcription factors may participate in DNMT3a gene activation (page 14, lines 20-22). Although OCT1 may also regulate the expression of other genes besides DNMT3a, such a study is out of the scope of this manuscript. We have another project to focus on the role of DRG OCT1 in neuropathic pain currently undergoing in the lab.

11. Often in the manuscript the data is presented normalized to contra. I think that the authors are missing the opportunity to display the contra data which could be very interesting. Why not normalizing to sham?

Because we did not know whether sham surgery itself affected the expression of the targeted genes (e.g., *DNMT3a* mRNA), we used the naïve animals as the control. For the data from RT-PCR experiments, ratios of ipsilateral-side mRNA levels to contralateral-side mRNA levels at the different time points after SNL or sham surgery were calculated using the ΔCt method ($2^{-\Delta\Delta\text{Ct}}$) after each was normalized to housekeeping genes such as *Gapdh* or *Tuba-1a*, because SNL was demonstrated not to alter the expression of our genes of interest on the contralateral side (from pilot work). For Western blot data, the relative density values from other time points after SNL or sham surgery were determined by dividing the optical density values from these groups by the value of the control/naïve group after each was normalized to the corresponding housekeeping proteins. The changes in the protein expression on BOTH ipsilateral and contralateral sides after SNL or sham surgery were examined.

12. The style is sometimes unclear and there are quite few typos. E.g.: in the "Figure Legends", the Tukey test is referred to throughout as the "Turkey test".

We have carefully gone through the manuscript and made sure there are no typos and have used the same style.

13. Fig.2a and 2b should be on the same scale (Y-axis).

Fig. 2a and Fig. 2b have been changed on the same scale in the Y-axis.

Response to the Reviewer #2 comments:

The manuscript by Zhao et al. reports novel results demonstrating that nerve injury-induced upregulation of the methyltransferase DNMT3a is specifically involved in the downregulation of the DRG potassium channel Kcna2, which induces DRG hyperexcitability and neuropathic pain. The authors make a very strong case for this conclusion by measuring multiple parameters (RNA, protein, function, behavior) and assessing several complementary conditions (sham, SNL, shRNA, fl/fl + Cre, overexpression, etc.). Furthermore, they characterized the injury-induced upregulation of OCT1, the transcription factor that regulates the DNMT3a promoter. In the end, the data strongly suggest that OCT1 promotes DNMT3a expression, which in turn selectively methylates the Kcna2 promoter and this, through an unknown mechanism, silences expression of the potassium channel. To the best of my knowledge, this is the first study that demonstrates in detail how a specific epigenetic modification plays a key role in the development of neuropathic pain through its effect on the expression of a particular potassium channel previously implicated in this malady.

Overall, the manuscript is well written, the approach is extensive and thorough, the quality of the experiments is excellent and the conclusions follow logically from the experimental results. Also, the methods provide sufficient experimental detail for possible

replication. There are, however, several important issues that the authors need to address to clarify the results, their interpretation and the conclusions.

Major Comments

1) The authors conclude that DNMT3a upregulation selectively downregulates Kcna2 (compared to Kcna1 and Kcna4) and that this dysregulation is mainly responsible for the increased excitability of large- and medium-size DRG neurons. However, it is clear that SNL also downregulates the Kcna1 and Kcna4 transcripts in the DRG (Fig. 6a/c) and that the largest change induced by the DNMT3a treatment was to increase the excitability of small-size DRG neurons (Fig. 8i). This is intriguing because Kcna2 is poorly expressed in small-size DRG neurons. An expanded critical discussion of this interesting and important observation would add impact to this study (including possible explanations).

This concern raised by the reviewer has been addressed in the Discussion part (page 16, lines 22-23; page 17, lines 1-8). We propose that, in addition to Kcna2, DNMT3a may also regulate the expression of other genes that are expressed in small DRG neurons and affect their excitability, although DNMT3a does not regulate the expression of Kcna1 and Kcna4 in small DRG neurons.

2) Previously, the authors reported that, following nerve injury, upregulation of the Kcna2 long non-coding antisense RNA attenuates expression of Kcna2. However, the exact mechanism underlying this outcome is not fully understood. Toward solving this mechanism, the authors refer to a previous study, which showed that the long non-coding antisense RNA Tsix specifically binds to DNMT3a to mediate de novo DNA methylation at the Xist promoter and induce repression of Xist. Although this is tantalizing, the authors only speculate that a similar mechanism might underlie the silencing effect of the Kcna2 long non-coding antisense RNA. Unless the authors have experimental evidence or there is established generality for the proposed mechanism, I suggest to tone down/delete this conclusion.

We agree with the reviewer. This part of the discussion has been re-written (page 16, lines 5-11).

Minor Comments

1. Following SNL, does the OCT1 mRNA increase in the DRG?

Yes, our pilot data showed that SNL increased OCT1 mRNA in the injured DRG. These data will be present in another manuscript.

2. Line 35: It would be more accurate to say "attenuating Kcna2 expression" than "silencing DRG Kcna2".

Thanks for the reviewer's suggestion. We have corrected it (page 2, lines 14-15).

3. Line 26: insert "the" between "of" and "dorsal root ganglion".

Thanks for the correction (page 2, lines 4).

4. Please consider revising the sentence running from line 38 to line 40.

We have re-written this sentence (page 3, lines 2-3).

5. Throughout the figure legends "Tukey" is misspelled ("Turkey"). Please revise.

We have corrected this typos.

Response to the Reviewer #3 comments:

The manuscript by Zhao et al. explores the role of DNMT3a in regulating voltage-gated potassium channels in the DRG of neuropathic rodents and the subsequent experience of pain. The main subject of these investigations, Kv1.2, and related K-channels are very well established to contribute to allodynia in nerve injury models. What is novel here is the establishment of a pathway leading from nerve injury to down-regulation of one target of DNMT3a. There are other genes known to be regulated by methylation in peripheral, DRG and spinal tissue in various pain models. This is a particularly strong story for one gene. Overall, the experimental approach is solid and multidisciplinary, and the methodology rigorous.

General comments:

1. Reflexive measures of nociception can no longer be considered to be adequate tests of the pain-like state that may be present in this and other pain models. Especially for a higher impact journal, tests of spontaneous pain such as CPP, vocalization, etc. or functional effects of the injury, e.g. gait, should be included. Sensitization alone is inadequate for the purpose of translation.

We agree with the reviewer. CCP test was carried out and the effect of DNMT3a shRNA on SNL-induced spontaneous pain was examined. New data have been added (page 8, lines 4-11. Fig. 4f; Supplemental Fig. 2i)

2. The authors measure the effects of DNMT3a on one gene, but there is no reason to think that many genes are not under the same enzyme's control, including others that might contribute to pain. One of the main characteristics of epigenetic processes is that they generally are not single-gene selective.

The reviewer is right. Based on our electrophysiological recording in small DRG neurons, DNMT3a may be involved in nerve injury-induced silencing of other Kv channels in small DRG neurons, given that DRG DNMT3a overexpression decreased total Kv currents and increased the excitability in small DRG neurons (in which Kcna2 is poorly expressed). Additionally, the participation of DNMT3a in nerve injury-induced

downregulation of non-Kv channel genes in the injured DRG could not be ruled out. These points have been discussed on page 16, lines 22-23; page 17, lines 1-8).

3. The authors make use of decitabine/zebularine/5azadC. Mechanistically, this drug requires cell proliferation in the tissues/cells of interest and should only target DNMT1. Why wasn't another drug used? (For example, RG101 is more specific and specifically targets DNMTa/b). The inclusion of the knockdown data helped narrow down the function to DNMT3A but, in general, decitabine should not be working on DNMT3a.

We appreciate the reviewer's points. To avoid the confusion, the decitabine data have been deleted. The DNMT3a knockdown and overexpression data from rats and mice should be enough to support our conclusion.

4. The authors use mouse and rat SNL models throughout the manuscript. Could the authors comment on whether they observed any species difference in instances where both species were examined for the same parameters?

We carried out two strategies, shRNA and Cre. Both approaches are complementary to each other. Rats intrinsically are less sensitive to the same intensity of either thermal or cold stimuli compared to mice. For thermal testing, we adjust the intensity of the thermal beam between species, such that a naïve animal will respond between 10 and 15 seconds (Fig. 4 d, j). For cold testing, no intensity adjustment is made as the cold plate is set at 0 degrees C, for this reason, the naïve rats have a longer latency than mice as depicted in Fig. 4e. However, it should be noted that there was no significant difference in the degree to which DNMT3a knockdown affected behavior (Fig 4d, e, j, k).

5. The discussion is simply a repetition of the "results" section and offers no real insight into the significance of the presented data or their fit with the current state of knowledge. Furthermore, both the introduction and discussion sections are missing key pieces of published literature from the field of pain epigenetics.

We have re-written the Discussion part and added some key references relevant to pain epigenetics in the Introduction part (page 3, lines 20-22).

6. Line 63/64 and 278/279: The claim that "DNMT3A is a new pain target" is overstated. DNMTs are widely and ubiquitously expressed, and while it may be tempting to over-interpret the findings and claim "therapeutic potential", the findings presented in the manuscript offer very limited data for such broad statements. Chemotherapeutics targeting methyltransferases have very severe side effects.

We have re-written the sentences to avoid any interpretation of exaggeration. (page 4, lines 4-5; page 17, lines 19-23; page 18, lines 1-2).

Figures:

There are currently 8 (very busy) figures that are frustrating to go through. Perhaps the authors could consider revising the content and organization of their panels and/or moving some of these data to supplemental information. For instance, figure 1 does not add much to the main manuscript and would be an ideal candidate for supplementary data.

All figures have been re-organized and some data moved into the supplemental information. Fig. 1 shows the cellular distribution of DNMT3a in the DRG, which has not been reported before. Given that this is basic information for the readers to understand the function of DNMT3a in neuropathic pain, we suggest to keep Fig. 1 in the main text.

Figure 2: The DNMT3b bands, particularly in the sham group, are very "fuzzy" and I have doubts whether they could be quantified with any level of reliability.

The level of DNMT3b expression is relatively low under normal conditions. We had to increase the loading samples and/or to extend the exposure time. New blots clearly displayed the DNMT3b bands.

Figure 4: This figure is extremely confusing due to the placement of the symbol legends. For each of the 4 rows of data, please have a dedicated space for all the symbol legends so that the reader is spared going back and forth through the panels to decipher the different symbols.

Fig. 4 has been re-organized based on the reviewers' suggestion.

Reviewers' comments:

Reviewer #1 (Remarks to the Author):

The authors have carried out a large amount of supplementary work and answered most of my queries. However, few points must be further discussed.

1. Concerning the quantification of the Westerns, I still strongly disagree with the control group being given the value of 100%, without any SEM. If we cannot evaluate the variability within the control group, then there is no way to evaluate whether the control group is different from the other groups. A correct way to do this would be to give to the control group average the value 100%, but then to recalculate all individual control group value as a ratio. Then you will be able to calculate the SEM.
2. Again regarding the westerns: I had mentioned that the quality of the band for the loading control H3 did not seem good enough to quantify individual wells. As a result, the authors have provided new images for the H3 loading, completely different from the original ones, but have left all other results unchanged! If you have provided new images, these must be quantified and new results must be presented.
3. Concerning Fig.4S, the experiment on promoter activity: could the authors clarify what is plotted on the Y-axis (relative activity)?
4. I believe that in Fig.7, the Fig.7F is lacking the letter F.
5. L.329: instead of "unobserved Kcan2 gene regions", maybe it would be more correct to say: "Kcan2 gene regions that were not studied".
6. Fig.1: thank you for providing a picture of satellite cells with labelled nuclei; however a high power magnification would be required to reach any conclusion.

Reviewer #2 (Remarks to the Author):

Zhao et al. have satisfactorily responded to my concerns/questions. A few remaining minor issues are listed below:

- 1) Abstract, line 26: Based on a previous suggestion to simply add "the" to a sentence, the authors modified by adding "the" AND "neurons". If "neurons" is kept, which should be good in the context of the sentence, please delete "the".
- 2) Page 12, line 266: Please replace "depolarized" for "increased", which is a more explicit description of the change in membrane potential.
- 3) Fig. 4 c, d, e, i, j, k: Symbol labeling for these panels is a bit confusing because the definition of a given symbol is above the graphs that use all pertinent symbols. Having all symbol definitions together would make the reading of these panels easier.

4) Fig. 5 c, d, e: same problem mentioned above.

Reviewer #3 (Remarks to the Author):

The authors have provided a re-submission of their manuscript, "Contribution of DNMT3a to neuropathic pain genesis by downregulating Kcna2 in primary afferent neurons." This is a very large set of studies focused on the role of Dnmt3a and its possible role in controlling neuropathic changes in rodents. Again, the strength of the study is in the number of overlapping techniques used as well as the inclusion of two species and to types of pain models. The authors were responsive to many of the concerns raised during the first round of review.

Some concerns do remain over the quality of some of the Western blot images and breadth of the Discussion. In addition, the presentation of the results becomes a little confusing in some areas with multiple species, time courses, DRG levels and measures. After re-reading the report I feel that this is a very complete analysis of a specific epigenetic issue of high value to the field of pain medicine and more moderate value as a sentinel set of observations to biomedical science in general.

The point-to point responses to the reviewers' comments.

Response to the Reviewer #1 comments:

1. Concerning the quantification of the Westerns, I still strongly disagree with the control group being given the value of 100%, without any SEM. If we cannot evaluate the variability within the control group, then there is no way to evaluate whether the control group is different from the other groups. A correct way to do this would be to give to the control group average the value 100%, but then to recalculate all individual control group value as a ratio. Then you will be able to calculate the SEM.

According to the reviewer's suggestion, we recalculated the values of all groups including the control groups. Now the control groups have the SEMs. All data have been reanalyzed. New outcomes are consistent with the previous conclusion.

It should be pointed out that some control groups do not have the SEMs, given that the values in these control groups are the same (for example in Fig. 5f).

2. Again regarding the westerns: I had mentioned that the quality of the band for the loading control H3 did not seem good enough to quantify individual wells. As a result, the authors have provided new images for the H3 loading, completely different from the original ones, but have left all other results unchanged! If you have provided new images, these must be quantified and new results must be presented.

We are very sorry about this mistake. Each Western blot experiment has been repeated for at least 3 times. In the revised version, we show new Western blot data for the target protein and the corresponding H3.

3. Concerning Fig.4S, the experiment on promoter activity: could the authors clarify what is plotted on the Y-axis (relative activity)?

To examine the promoter activity, the second reporter is often used as an internal control for normalization. We used *renilla* luciferase reporter gene as an internal control. We co-transfected pGL3 vector (containing *firefly* luciferase reporter gene) and *renilla* luciferase vector into cultured cells and measured the luciferase activity using the Dual-Luciferase reporter assay system. Thus, the "relative activity" is the ratio of *firefly* luciferase activity to *renilla* luciferase activity. We have clarified this point in the Methods Part (page 26, line 22; page 27, lines 2-3; lines 7-8).

4. I believe that in Fig.7, the Fig.7F is lacking the letter F.

In Fig. 7, Fig 7 e showed the percentages of methylation at -457 and -444 CpG sites of the *Kcna2* gene in the injured DRG on day 7 post-SNL or sham surgery from five groups. Left: Representation of a single cloned allele per group at -457 CpG site. Right: Statistical analysis at -457 and -444 CpG sites. There is no Fig. 7f (page 11, line 17; page 37, lines 21-22).

5. L.329: instead of “unobserved *Kcna2* gene regions”, maybe it would be more correct to say: “*Kcna2* gene regions that were not studied”.

This statement has been corrected as suggested (page 15, line 18).

6. Fig.1: thank you for providing a picture of satellite cells with labelled nuclei; however a high power magnification would be required to reach any conclusion.

A higher power magnification is now provided (Fig. 1b).

Reviewer #2 (Remarks to the Author):

Zhao et al. have satisfactorily responded to my concerns/questions. A few remaining minor issues are listed below:

1) Abstract, line 26: Based on a previous suggestion to simply add “the” to a sentence, the authors modified by adding “the” AND “neurons”. If “neurons” is kept, which should be good in the context of the sentence, please delete “the”.

We deleted “the” as suggested.

2) Page 12, line 266: Please replace “depolarized” for “increased”, which is a more explicit description of the change in membrane potential.

We have made the correction as suggested (page 12, line 23).

3) Fig. 4 c, d, e, i, j, k: Symbol labeling for these panels is a bit confusing because the definition of a given symbol is above the graphs that use all pertinent symbols. Having all symbol definitions together would make the reading of these panels easier.

We have presented all symbol definitions together in Fig. 4c, d, e, i, j, k as suggested.

4) Fig. 5 c, d, e: same problem mentioned above.

We also did the same changes as described above in Fig. 5 c, d, e as suggested.

Reviewer #3 (Remarks to the Author):

Some concerns do remain over the quality of some of the Western blot images and breadth of the Discussion. In addition, the presentation of the results becomes a little confusing in some areas with multiple species, time courses, DRG levels and measures. After re-reading the report I feel that this is a very complete analysis of a specific epigenetic issue of high value to the field of pain medicine and more moderate value as a sentinel set of observations to biomedical science in general.

We thank the reviewer for his/her positive comments. As in the response to the first and second reviewers' concerns described above, the quality of the Western blot images has been improved. A more detailed discussion has been given. In the figure legends, we have clarified the animal species (mice vs rats), time courses, DRG levels, and measures.

REVIEWERS' COMMENTS:

Reviewer #1 (Remarks to the Author):

Zhao et al. have satisfactorily addressed my concerns.

The point-to point responses to the reviewers' comments.

Response to the Reviewer #1 comments:

Zhao et al. have satisfactorily addressed my concerns.

Thanks. We are happy that the reviewer satisfies our responses.